# Nuclear rupture in confined cell migration triggers nuclear actin polymerization to limit chromatin leakage

Christos Kamaras [ID] [1,4 ✉], Dennis Frank [ID] [1,4], Hong Wang [ID] [1,2], Friedel Drepper [ID] [3], Pitter F Huesgen [ID] [2,3] & Robert Grosse [ID] [1,2 ✉]

## Abstract

**Upon cell migration in confined space, such as during cancer metastasis, mechanical forces from the extracellular matrix act onto the nucleus leading to nuclear envelope (NE) rupture, chromatin leakage and genomic instability. Here we found that during confined migration, NE rupture triggers dynamic nuclear F-actin formation dependent on the formins DIAPH1 and DIAPH3. We show that DIAPH3 dynamically and transiently relocates to the nucleus upon NE rupture. Interfering with DIAPH1/3 or with nuclear actin polymerization resulted in nuclear instability during confined migration. Notably, nuclear formin activity or actin assembly limit NE rupture-induced chromatin leakage. Similarly, silencing of Ataxia Telangiectasia and Rad3-related protein (ATR) reduced NE rupture-triggered nuclear F-actin assembly and increased chromatin leakage. Consistent with this, ATR promotes the phosphorylation of DIAPH3 at S1072 adjacent to its auto-regulatory domain to promote nuclear actin polymerization. Using atomic force microscopy, we found that nuclear actin assembly or nuclear DIAPH3 activity promotes nuclear stiffness in an ATR-dependent manner. Thus, our study identifies an ATR-formin module that regulates nuclear mechanical properties through induction of intranuclear actin scaffolding.**

**Keywords** Nuclear Actin; Nuclear Mechanics; Cancer Cell Invasion
**Subject Categories** Cell Adhesion, Polarity & Cytoskeleton; Chromatin, Transcription & Genomics

## Introduction

For cancer cells to form metastatic lesions to secondary sites, they have to disseminate from the primary tumor to invade the surrounding stroma in a process called local invasion (Chaffer and Weinberg, 2011). During this process, cells encounter a dense extracellular matrix (ECM) network forming confined spaces through which they must pass (Yamada and Sixt, 2019). The nucleus is the largest cellular organelle, hence, posing a rate-limiting step during invasive migration (Wolf et al, 2013; Davidson et al, 2014). During confined migration, the nucleus is exposed to mechanical forces acting on the nuclear envelope (NE) (Harada et al, 2014), which in turn frequently leads to NE rupture (Vargas et al, 2012) (Raab et al, 2016; Denais et al, 2016). As a consequence, genomic DNA leaks into the cytoplasm, leading to activation of inflammatory pathways (Frittoli et al, 2023) as well as to significant DNA damage (Nader et al, 2021). The kinases ataxia telangiectasia and Rad3-related protein (ATR) and ataxia-telangiectasia mutated protein (ATM) are the master regulators of the DNA damage response and ensure the integrity of the mammalian genome in eukaryotes (Awasthi et al, 2015). In the case of ATM, mechanical activation leads to rearrangements of the actin cytoskeleton and overall increased cellular stiffness, while nuclear stiffness remains unaffected (Bastianello et al, 2023). ATR, in turn, has been reported to translocate to the NE in response to mechanical stimuli (Kumar et al, 2014) and cells depleted of ATR display softer and more fragile nuclei (Kidiyoor et al, 2020). Furthermore, ATR has been linked to DNA damage-induced NE rupture through phosphorylation of Lamin-A/C (Kovacs et al, 2023), while loss of ATR in cancer cells impedes the progression of tumors, thus rendering ATR as an attractive drug target (Kidiyoor et al, 2016). However, the mechanism by which ATR controls nuclear mechanics is not well understood.

The formin family of actin nucleators functions as potent factors for actin polymerization by incorporating actin monomers in the barbed end, resulting in the formation of linear actin filaments (Faix and Grosse, 2006), and their upregulation has been linked to various types of cancer often associated with poor clinical prognosis (DeWard et al, 2010). Rho-GTPase binding or phosphorylation events have been reported to regulate the activity of some formins to stimulate actin remodeling (Wang et al, 2015; Staus et al, 2011). Interestingly, some formins are also present in the nucleus (Miki et al, 2009; Baarlink et al, 2013). Indeed, diaphanous-related formin 1 (DIAPH1), disheveled associated activator of morphogenesis 2 (DAAM2), and inverted formin 2 (INF2) have been shown to

[1]Institute of Experimental and Clinical Pharmacology and Toxicology, Medical Faculty, University of Freiburg, 79104 Freiburg, Germany. [2]CIBSS Centre for Integrative Biological Signaling Studies, University of Freiburg, 79104 Freiburg, Germany. [3]Institute of Biology II, University of Freiburg, 79104 Freiburg, Germany. [4]These authors contributed equally: Christos Kamaras, Dennis Frank. ✉E-mail: christos.kamaras@pharmakol.uni-freiburg.de; robert.grosse@pharmakol.uni-freiburg.de

promote nuclear actin polymerization in response to extracellular stimuli (Ulferts et al, 2024; Ulferts and Grosse, 2024; Knerr et al, 2023).

Actin exists in the nucleus in both monomeric (G-actin) and filamentous (F-actin) forms. The nuclear actin pool is tightly controlled through active import and export mechanisms. Import of actin into the nucleus in complex with cofilin is facilitated by importin-9 (Dopie et al, 2012), while nuclear export of actin in complex with profilin is mediated by exportin-6 (Stuven et al, 2003). While G-actin is an integral part of chromatin remodeling (Kapoor and Shen, 2014), F-actin in the nucleus has remained undetected for a long time. However, with the development of new tools to visualize nuclear actin in living cells (Melak et al, 2017), the dynamic nature of nuclear actin assembly has been linked to multiple nuclear functions (Ulferts et al, 2024). Transient and dynamic nuclear actin polymerization has been implicated in the regulation of serum response factor (SRF) (Baarlink et al, 2013), androgen signaling (Knerr et al, 2023), transcription (Baarlink et al, 2013; Plessner et al, 2015; Knerr et al, 2023; Wei et al, 2020), chromatin reorganization (Mahmood et al, 2021; Baarlink et al, 2017; Wang et al, 2019), DNA damage and replication stress repair (Wang et al, 2019; Baarlink et al, 2017; Caridi et al, 2018; Schrank et al, 2018; Krippner et al, 2020; Sen et al, 2024), and in response to osmotic stress in an ATR-dependent manner (Chatzifrangkeskou et al, 2025). Nevertheless, the potential role for nuclear actin assembly during cell migration has not been investigated.

# Results

## NE rupture precedes the formation of dynamic nuclear F-actin structures

In order to observe single cells migrating under 3D confinement we used microfluidic devices that consisted of microchannel constrictions of 3 or 8 μm (Fig. 1A). We employed highly invasive fibrosarcoma cells (HT1080), that stably express the nuclear actin-chromobody (nAC), a cameloid-derived nanobody that specifically detects endogenous actin in the nucleus of living cells (Plessner et al, 2015). HT1080 cell migration through microchannels of either 8 μm or 3 μm narrow constrictions was monitored by high-resolution live-cell imaging with enhanced spatial resolution Airyscan processing. While moderate confinement via 8 μm microchannels did not result in nuclear actin assembly, migration through 3 μm microchannels induced the formation of short and dynamic nuclear F-actin structures (Fig. 1A; Movie EV1). Furthermore, polymerization of nuclear actin during migration through 3 μm coincided with a steep decrease of nAC-GFP intensity in the nucleus (Fig. 1A) caused by leakage of the probe from the nucleus into the cytosol due to NE rupture. The majority of the cells migrating through 3 μm channels experienced NE rupture, as observed by nAC-GFP nuclear leakage (Fig. 1B,C). Overexpression of a dominant-negative KASH mutant, which disrupts the connections between the cytoskeleton and the NE (Lombardi et al, 2011), did not affect the formation of detectable nuclear F-actin structures (Fig. EV1A,B; Movie EV2), suggesting that nuclear actin polymerization during confined migration through 3 μm is not mediated by the linker of nucleoskeleton and cytoskeleton (LINC) complex. We also imaged non nAC-

expressing cells using FastActX, a novel live-cell actin imaging dye which labels highly dynamic actin filaments (Nasufovic et al, 2025). Upon NE rupture, evidenced by NLS-BFP leakage, we observed dynamic nuclear F-actin formation, showing that this process occurs also in the absence of genetically encoded actin probes (Fig. EV1C; Movie EV3). Furthermore, nuclear actin assembly after NE rupture could be co-visualized by FastActX and nAC-GFP (Fig. EV1D; Movie EV4), suggesting that they can label identical actin structures.

To more directly assess NE rupture, we expressed a catalytically inactive cyclic-GAMP synthase probe (icGAS), a widely used NE rupture marker (Raab et al, 2016; Nader et al, 2021; Frittoli et al, 2023), which detects double-stranded DNA leaking from the nucleus to the cytoplasm. Using this probe, we could confirm that the vast majority of the cells migrating through 3 μm exhibited NE rupture as monitored by perinuclear accumulation of icGAS. Strikingly, most of the icGAS-positive cells also displayed the formation of nuclear F-actin (Fig. 1D–F; Movie EV4) (Denais et al, 2016). By exploiting high temporal resolution during live-cell imaging, we observed that NE rupture preceded the formation of detectable nuclear F-actin structures with a short time frame of $80.09 \pm 25.46$ s (Fig. 1G; Movie EV5), showing that confinement-induced NE rupture precedes nuclear actin polymerization.

## NE rupture-mediated nuclear actin assembly is mediated by formins and triggers Diaph3 nuclear localization

We next aimed to determine the actin assembly factor responsible for nuclear actin polymerization upon NE rupture. Treatment of cells with the specific Arp-2/3 complex inhibitor CK-666 had no effect on nuclear F-actin assembly (Fig. EV2A,B). We then performed siRNA experiments targeting formin family members previously implicated in nuclear actin polymerization (Baarlink et al, 2013; Wang et al, 2019).

Following depletion of DIAPH1 or DIAPH3, significantly fewer cells displayed nuclear actin filaments upon NE rupture. Double knockdown of both DIAPH1 and DIAPH3 had a more pronounced effect, indicating that the two formins may act in a synergistic manner. Depletion of DIAPH2 and INF2 had no significant effect on the formation of nuclear actin filaments in cells migrating through 3 μm channels (Figs. 2A,B and EV2C; Movie EV6). ROCK-mediated phosphorylation has been shown to play a role in DIAPH3 activation and subsequent actin polymerization (Staus et al, 2011). However, treatment of cells with a specific ROCK inhibitor had no effect on nuclear F-actin structures upon NE rupture (Fig. EV2D,E), suggesting the existence of another context-dependent activation of DIAPH3.

We then examined migrating HT1080 cells expressing GFP-Diaph3. Interestingly, in cells migrating through 3 μm microchannels, GFP-Diaph3 localized to the cytosol and to the perinuclear region but upon NE rupture rapidly and transiently translocated to the nucleus (Fig. 2C,D; Movie EV7). In contrast, another diaphanous-related formin, formin-like 2 (FMNL2), did not accumulate in the nucleus following NE rupture (Fig. EV2F–H; Movie EV8), indicative of a specific signal-mechanism for Diaph3 nuclear entry under these conditions. Thus, NE rupture triggers nuclear accumulation of the formin actin regulator Diaph3 in a highly temporal manner.

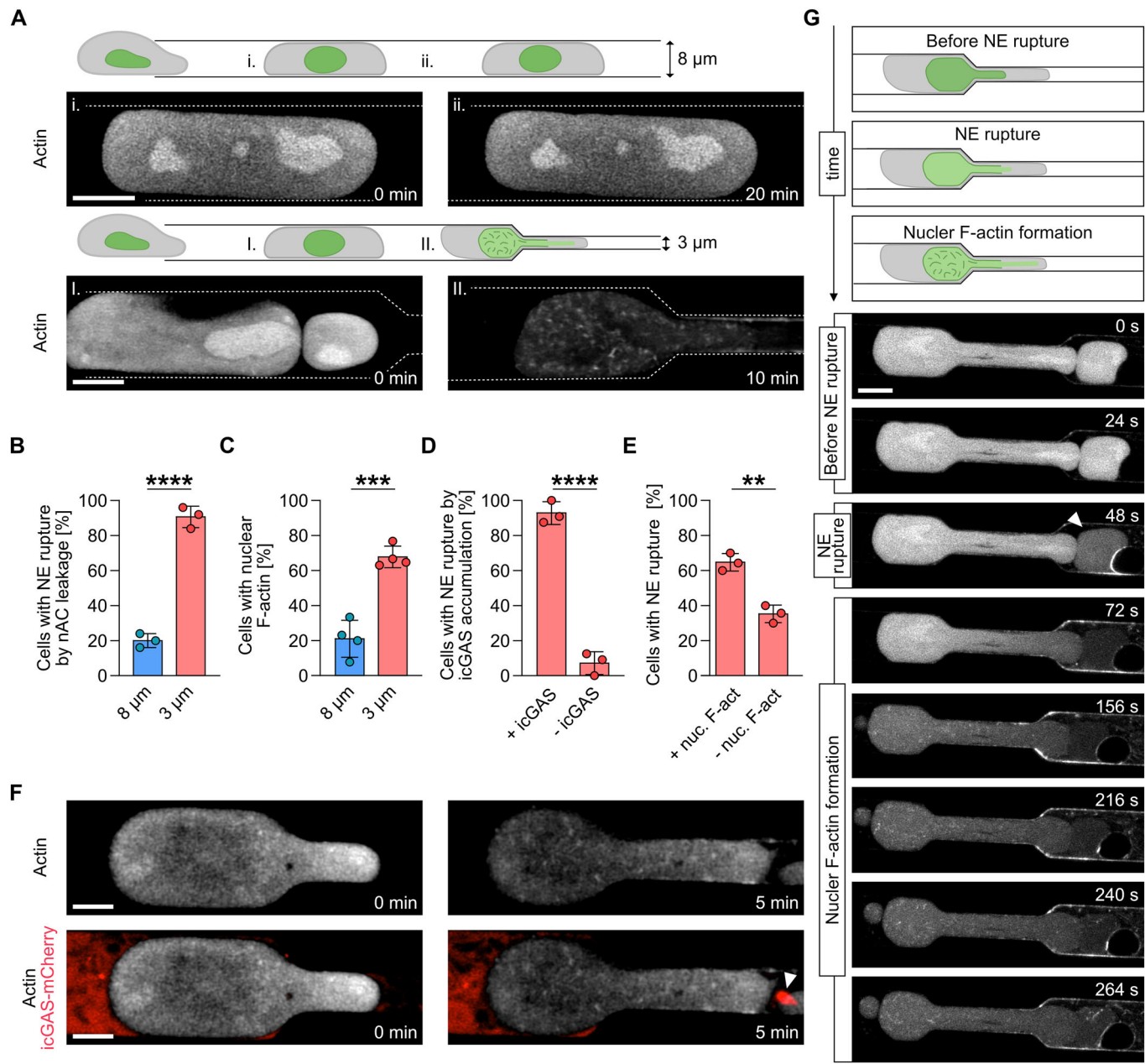

**Figure 1. Nuclear envelope rupture during confined migration triggers the formation of dynamic nuclear F-actin structures.**

(A) Schematic representation and image sequence of a HT1080 cell stably expressing nuclear Actin Chromobody (nAC-GFP) migrating through an 8 μm (top) or 3 μm microchannel (bottom) with schematic representations of image sequences in 8 μm (i, ii, I) and in 3 μm (II) microchannels. Dashed white lines indicate microchannel walls. Actin is visualized via nAC-GFP. Scale bar, 5 μm. (B) Percentage of HT1080-nAC-GFP cells displaying leakage during migration through 8 μm (blue) or 3 μm (red) microchannels. Data shown as mean ± s.d over three independent experiments per condition with $n = 25$ cells per experiment. Statistical analysis was performed by an unpaired parametric $t$-test. Exact $P$ value: 0.00007. (C) Percentage of HT1080 cells stably expressing nAC-GFP displaying nuclear F-actin formation during migration through 8 μm (blue) or 3 μm (red) microchannels. NE nuclear envelope. Data shown as mean ± s.d over four independent experiments per condition with $n \geq 9$ cells per experiment. Statistical analysis was performed by unpaired parametric $t$-test. Exact $P$ value: 0.0003. (D) Percentage of icGAS-positive cells after migration through 3 μm (red) microchannels. Data shown as mean ± s.d over three independent experiments per condition with $n \geq 22$ cells per experiment. Statistical analysis was performed by unpaired parametric $t$-test. Exact $P$ value: 0.00008. (E) Percentage of icGAS-mCherry-positive HT1080 cells that display nuclear F-actin formation after migration through 3 μm microchannels (red). Data shown as mean ± s.d over three independent experiments per condition with $n \geq 10$ cells per experiment. Statistical analysis was performed by unpaired parametric $t$-test. Exact $P$ value: 0.002. (F) Representative image sequences of an HT1080 cell stably expressing nAC-GFP (white) and icGAS-mCherry (red) migrating through a 3 μm microchannel. The NE rupture site is indicated by accumulation of icGAS-mCherry (white arrowhead). Actin is visualized via nAC-GFP. Scale bar, 5 μm. (G) Schematic representation (top) and representative image sequences (bottom) of an HT1080 cell stably expressing nAC-GFP migrating through a microchannel, before NE rupture, during NE rupture and during nuclear F-actin formation upon NE rupture. Scale bar, 5 μm. (B–E) Statistical analysis was performed by unpaired parametric $t$-test; **$P < 0.01$, *** $P < 0.001$, ****$P < 0.0001$. Source data are available online for this figure.

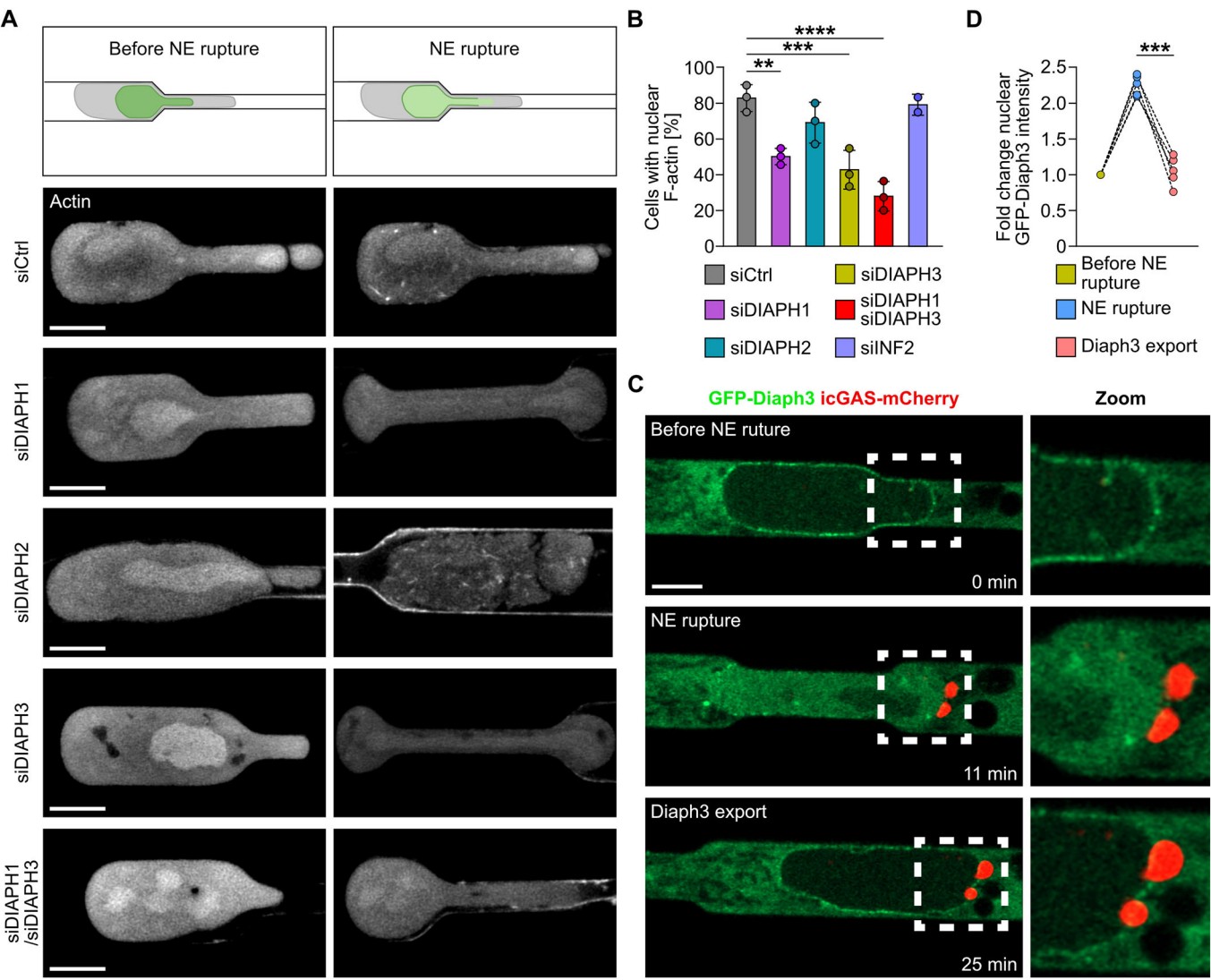

**Figure 2. DIAPH1 and DIAPH3 mediate nuclear actin polymerization upon NE rupture.**

(A) Top, schematic representation of a cell migrating through a 3 μm microchannel. Bottom, representative image sequences of HT1080 cells stably expressing nAC-GFP migrating through 3 μm microchannels. HT1080 cells are treated with control siRNA (siCtrl) or siRNA targeted against DIAPH1 (siDIAPH1), DIAPH2 (siDIAPH2), DIAPH3 (siDIAPH3) or DIAPH1 and DIAPH3 combined (siDIAPH1/siDIAPH3). Actin is visualized via nAC-GFP. Scale bar, 5 μm. (B) Percentage of HT1080 cells stably expressing nAC-GFP displaying nuclear F-actin formation after treatment with either siCtrl, siDIAPH1, siDIAPH2, siDIAPH3, siDIAPH1/siDIAPH3, or siINF2. Data shown as mean ± s.d. Three independent experiments per condition and two independent experiments for siINF2, $n \geq 9$ cells per experiment. Statistical analysis was performed by one-way analysis of variance (ANOVA). Exact P values from left to right: $P = 0.00296$, $P = 0.0006$, $P = 0.00003$. (C) Representative image sequences of an HT1080 cell transiently expressing GFP-Diaph3 (green) and icGAS-mCherry (red) migrating through a microchannel. Scale bar, 4 μm. (D) Quantification of nuclear GFP-Diaph3 intensity before NE rupture, during NE rupture and after Diaph3 export. Data shown as individual values from $n = 5$ cells. Statistical analysis was performed by paired parametric $t$-test; Exact P value: 0.00054. ns $P > 0.05$, **$P < 0.01$, ***$P < 0.001$, ****$P < 0.0001$. Source data are available online for this figure.

## DIAPH1/DIAPH3 and nuclear actin polymerization are required for nuclear integrity upon NE rupture

Next, we allowed HT1080 cells that stably co-express fluorescently tagged histone H2B (H2B-mCherry) and icGAS-GFP to migrate through 3-μm microchannels and imaged them over time via confocal microscopy. We noticed that cells depleted of DIAPH1 and DIAPH3 were more prone to nuclear fragmentation upon NE rupture as observed by H2B fluorescence (Fig. 3A–C; Movie EV9).

Overexpression of the non-polymerizable actin mutant R62D targeted to the nucleus (NLS-BFP-Actin-R62D) significantly increased the frequency of nuclear fragmentation after migration through 3 μm microchannels (Fig. 3A,D,E; Movie EV10), suggesting that nuclear actin assembly plays a role in nuclear integrity during confined migration.

Notably, impaired nuclear actin assembly either by depletion of DIAPH1 and DIAPH3 or by overexpression of NLS-BFP-Actin-R62D had no effect on the frequency of NE rupture events during confined migration (Fig. 3F–I; Movies EV11 and 12), supporting the notion that NE rupture is upstream of nuclear F-actin formation.

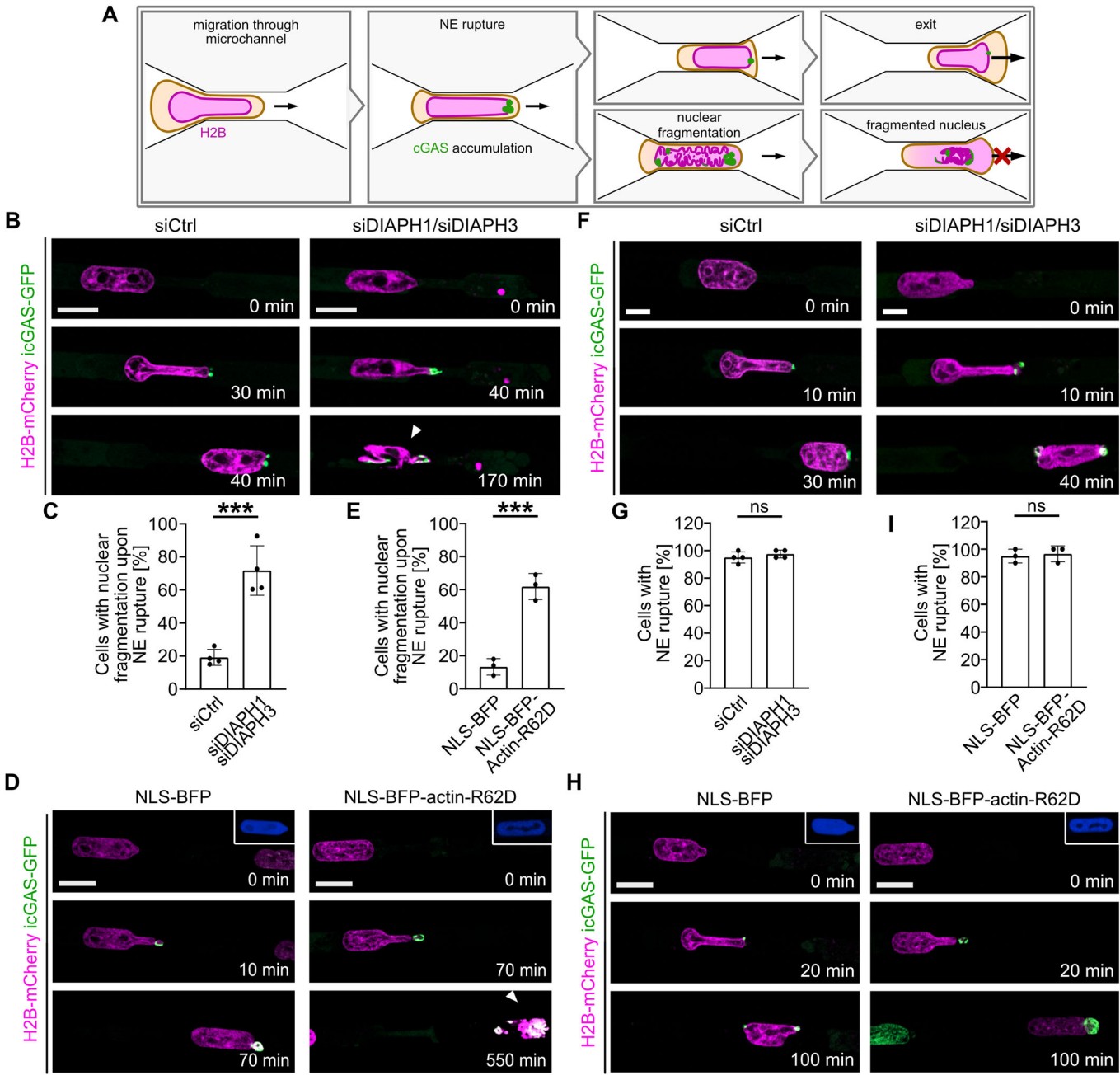

## DIAPH1/DIAPH3 and nuclear actin polymerization counteract chromatin leakage upon NE rupture

To study nuclear integrity in more detail and to investigate NE rupture-triggered chromatin leakage, we turned to 3D volume measurements using IMARIS as previously described (Knerr et al, 2023). Strikingly, silencing DIAPH1/DIAPH3 led to an increase of chromatin leakage as measured by icGAS as well as cytoplasmic DNA volume (Fig. 4A–C). Moreover, expression of NLS-Actin-R62D resulted in significantly more chromatin leaking (Figs. 4D–F and EV3A), suggesting that DIAPH1/DIAPH3 and nuclear actin assembly mediate genome protection during confined migration. These data suggest that formin-mediated nuclear actin polymerization is crucial to ensure nuclear integrity by

limiting chromatin leakage following confined migration-induced NE rupture.

## ATR is required for NE rupture-triggered nuclear actin polymerization

It has been previously demonstrated that ATR can be activated through mechanical inputs, ensuring nuclear integrity for interstitial migration (Kumar et al, 2014; Kidiyoor et al, 2020). This prompted us to test whether ATR, which becomes activated by nuclear squeezing or deformation, may be involved in nuclear actin polymerization upon confined migration-induced NE rupture. Interestingly, silencing of ATR abrogated the ability of cells to

**Figure 3. Nuclear actin polymerization is important for cell survival during confined cell migration.**

(A) Schematic representation of a cell migrating through a 3 μm microchannel. Nuclear deformation results in an NE rupture event as observed by perinuclear accumulation of icGAS, followed by either exit of the nucleus from the microchannel or nuclear fragmentation and halt of migration. (B) Representative image sequences of HT1080 cells stably expressing H2B-mCherry and icGAS-GFP migrating through 3 μm microchannels after treatment with either control siRNA (siCtrl) or siRNA against DIAPH1 and DIAPH3 (siDIAPH1/siDIAPH3), undergoing nuclear fragmentation. The white arrowhead indicates nuclear fragmentation. Scale bar, 10 μm. (C) Percentage of HT1080 cells stably expressing histone H2B (H2B-mCherry) and icGAS-GFP treated with siCtrl or siDIAPH1/siDIAPH3 displaying nuclear fragmentation upon NE rupture. Data shown as mean ± s.d. Four independent experiments per condition, $n \geq 27$ cells per experiment. Statistical analysis was performed with unpaired parametric $t$-test. Exact P value: 0.0005. (D) Image sequences of representative HT1080-H2B-mCherry/icGAS-GFP/NLS-BFP or NLS-BFP-Actin-R62D (blue) migrating through 3 μm microchannels, undergoing nuclear fragmentation. The white arrowhead indicates nuclear fragmentation. Scale bar, 10 μm. (E) Percentage of HT1080 cells stably expressing H2B-mCherry/icGAS-GFP/NLS-BFP or NLS-BFP-Actin-R62D displaying nuclear fragmentation upon NE rupture. Data shown as mean ± s.d. and are displayed in a scatter dot bar plot. Three independent experiments per condition with $n \geq 38$ cells per experiment. Statistical analysis was performed with an unpaired parametric $t$-test. Exact P value: 0.0008. (F) Representative image sequences of HT1080 cells expressing H2B-mCherry and icGAS-GFP migrating through 3 μm microchannels undergoing NE rupture, as observed by perinuclear accumulation of icGAS-GFP after being treated with either siCtrl or siDIAPH1/siDIAPH3. Scale bar, 10 μm. (G) Percentage of cells stably expressing H2B-mCherry/icGAS-GFP displaying NE rupture after being treated with either siCtrl or siDIAPH1/siDIAPH3. Data shown as mean ± s.d. Four independent experiments per condition with $n = 20$ cells per experiment. Statistical analysis was performed with the Mann–Whitney test. Exact P value: 0.6571. (H) Representative image sequences of HT1080-H2B-mCherry/icGAS-GFP expressing NLS-BFP or NLS-BFP-Actin-R62D migrating through 3 μm microchannels undergoing NE rupture, as observed by accumulation of icGAS-GFP. Scale bar, 10 μm. (I) Percentage of HT1080-H2B-mCherry/icGAS-GFP cells expressing NLS-BFP or NLS-BFP-Actin-R62D that undergo NE rupture as scored by perinuclear accumulation of icGAS. Data shown as mean ± s.d. Three independent experiments per condition with $n = 20$ cells per individual experiment. Statistical analysis was performed with the Mann–Whitney test. Exact P value: $P > 0.9999$. ***$P < 0.001$ and ns not significant. Source data are available online for this figure.

form NE rupture-triggered nuclear F-actin (Fig. 5A–C). Moreover, silencing of ATR phenocopied the increase of chromatin leakage, similarly to inhibition of DIAPH1/3 or nuclear actin assembly (Fig. 5D–F). This effect was specific since after validating the activity of the ATR inhibitors (Fig. EV4A), pharmacological inhibition of ATR but not of ATM impaired nuclear actin polymerization upon NE rupture (Fig. 6A,B). Interestingly, pharmacological targeting of ATR had no effect on the frequency of NE rupture events during migration through 3 μm microchannels (Fig. EV4B), indicating that NE rupture functions upstream in this process. These data demonstrate that ATR, but not ATM, promotes nuclear actin polymerization following an NE rupture. Notably, pharmacological inhibition of Chk1 with LY2603618 had no effect on nuclear F-actin formation (Figs. EV4C,D), suggesting that ATR controls formin activity independently of the canonical ATR-DNA damage response pathway.

## ATR regulates DIAPH3 phosphorylation to promote nuclear actin polymerization

In order to test whether ATR could be regulating the activity of DIAPH3 via phosphorylation, we performed liquid chromatography-mass spectrometry (LC-MS) phospho-peptide analysis on pulled-down myc-NLS-BFP-Diaph3. We aimed to identify potential phosphorylation sites upon ATR activation and we observed a single phosphorylated serine residue S1072 with a localization confidence of >99% that is part of a S-P motif adjacent to the diaphanous-autoregulatory-domain (DAD) (Fig. 7A) (Li and Higgs, 2005). Since the S-P motif appears to be conserved between the human DIAPH3 (UniProt ID: Q9NSV4) and the mouse orthologue Diaph3 (UniProt ID: Q9Z207), we generated a phospho-mimetic derivative Diaph3-S1072D, and a phospho-dead form Diaph3-S1072A (Fig. 7B) for nuclear targeting. In order to validate our LC-MS phospho-peptide analysis, we performed immunoprecipitation (IP) experiments on myc-NLS-BFP-Diaph3-wt or myc-NLS-BFP-Diaph3-S1072A as a control, following hydroxyurea (HU) treatment in the presence or absence of the ATR inhibitor ETP-46464. Using anti-phosphoserine antibodies,

we observed a significant increase in nuclear Diaph3 phosphorylation upon HU that was sensitive to ATR inhibition (Fig. EV4E,F). No increase in phosphorylation was detected with Diaph3-S1072A (Fig. EV4G,H), suggesting that ATR-mediated phosphorylation of Diaph3 requires the residue Ser1072. Overexpression of NLS-Diaph3-wt or NLS-Diaph3-S1072D induced nuclear actin polymerization even in the absence of NE rupture, while NLS-Diaph3-S1072A failed to form nuclear F-actin (Fig. 7C,D). We then pharmacologically inhibited ATR using ETP-46464, which significantly reduced nuclear actin assembly by NLS-Diaph3-wt but not by the phospho-mimetic NLS-Diaph3-S1072D (Fig. 7E,F), arguing that ATR regulates phosphorylation and activation of DIAPH3 at S1072 upon NE rupture.

## ATR-formin regulation and nuclear actin polymerization promote nuclear stiffness

We then sought to determine whether nuclear actin polymerization directly affects the mechanical properties of the nucleus. Nuclei of HT1080 cells transiently expressing NLS-mScarlet were probed using atomic force microscopy (AFM). Force-distance curves were recorded with 5 nN applied force, and nuclear deformation was monitored to ensure proper probing of the nuclei (Fig. 8A). Overexpression of NLS-Actin or the F-actin-promoting NLS-Actin-S14C (Posern et al, 2002) readily and robustly induced nuclear actin polymerization (Fig. 8B). Interestingly, we observed that NLS-actin as well as NLS-Actin-S14C resulted in significantly stiffer nuclei in comparison to NLS-Actin-R62D expressing cells (Fig. 8C,D). This demonstrates that nuclear F-actin can act as a scaffold to support nuclear stiffness and hence its mechanical properties. Consequently, we investigated whether Diaph3 regulates nuclear stiffness through actin polymerization in an ATR-dependent manner. Thus, we overexpressed the phospho-dead or phospho-mimetic NLS-Diaph3 constructs in HT1080 cells. We probed nuclei in the same dish before and after treatment with ETP-46464 and quantified the change in nuclear stiffness. Interestingly, cells expressing either NLS-Diaph3-wt or phospho-mimetic NLS-Diaph3-S1072D displayed significantly stiffer nuclei

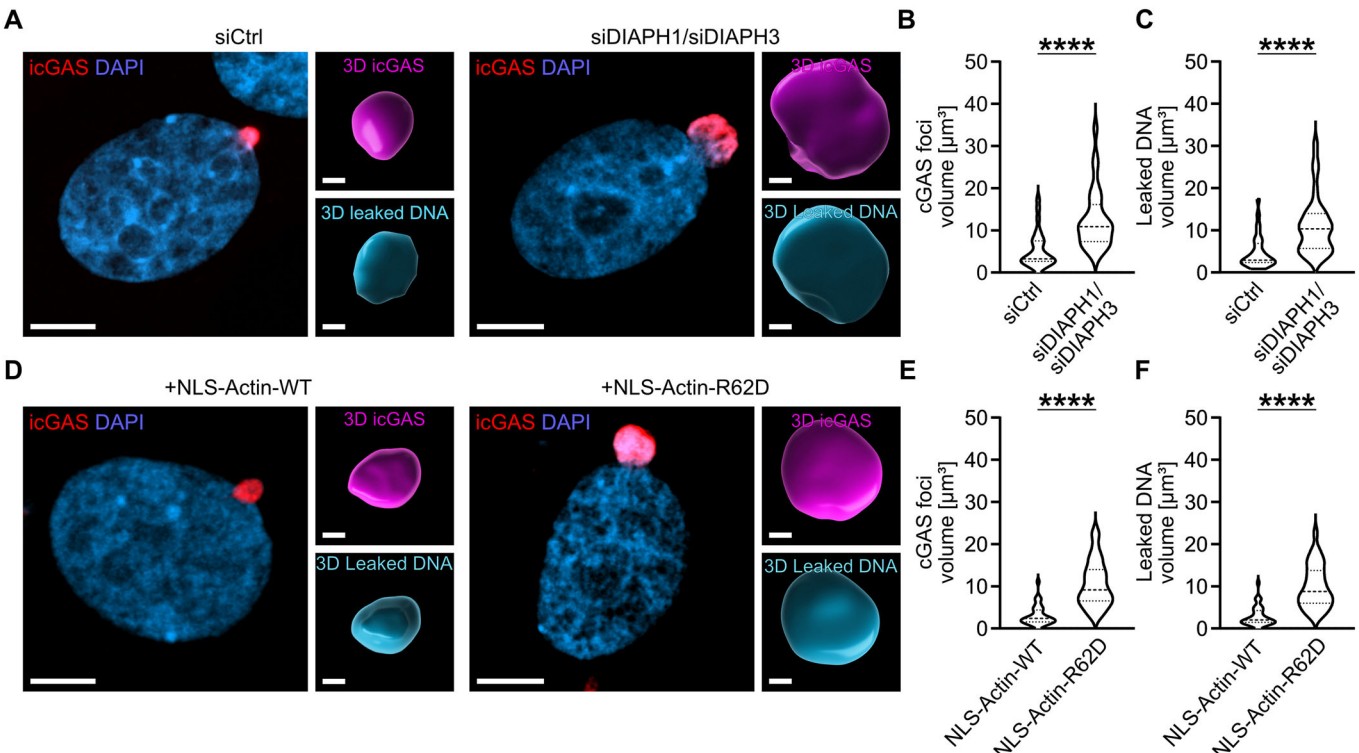

**Figure 4. Nuclear actin polymerization maintains nuclear integrity during confined cell migration.**

(A) Representative immunofluorescence (IF) images of HT1080 cells stably expressing icGAS-mCherry after being treated with either siCtrl or siDIAPH1/siDIAPH3. 3D-rendered images of icGAS-mCherry foci (magenta) and leaked chromatin (blue). Scale bar, 5 μm. Scale bar for 3D-rendered icGAS and 3D leaked DNA, 1 μm. (B) Quantification of 3D-rendered icGAS-GFP foci volume from (A). (C) Quantification of 3D-rendered DAPI volume in the cytoplasm from (A). Data from $n \geq 9$ cells were displayed in a violin plot with median and quartiles in three independent experiments. Statistical analysis was performed with Mann–Whitney test. Exact $P$ value from (B): $7.0 \times 10^{-7}$. Exact $P$ value from (C): $4.0 \times 10^{-6}$. Images are shown as maximum z-projections. Scale bar, 5 μm. (D) IF images of HT1080 cells stably expressing icGAS-mCherry and transiently expressing either NLS-Actin-WT or NLS-Actin-R62D. The expression of NLS-Actin-WT or NLS-Actin-R62D for these representative cells is shown in Fig. EV3A. 3D-rendered images of icGAS-mCherry foci (magenta) and leaked chromatin (blue). Scale bar, 5 μm. Scale bar for 3D-rendered icGAS and 3D leaked DNA, 1 μm. (E) Quantification of 3D-rendered icGAS-mCherry foci volume from (D). (F) Quantification of 3D-rendered DAPI volume in the cytoplasm from (D). Data from $n = 12$ cells were displayed in a violin plot with median and quartiles in three independent experiments. Statistical analysis was performed with Mann–Whitney test. Exact $P$ value from (E): $5.0 \times 10^{-12}$. Exact $P$ value from (F): $9.0 \times 10^{-12}$. Images are shown as maximum z-projections. Scale bar, 5 μm. ****$P < 0.0001$. Source data are available online for this figure.

in comparison to the phospho-dead NLS-Diaph3-S1072A expressing cells and control cells expressing NLS-mScarlet (Fig. 9A). Moreover, pharmacological inhibition of ATR in cells expressing NLS-Diaph3-wt decreased nuclear stiffness, while cells expressing NLS-Diaph3-S1072A or the constitutively active formin NLS-Diaph3-S1072D remained unresponsive to ATR inhibition (Fig. 9B,C). These data demonstrate that ATR kinase can influence nuclear rigidity by regulating formin-mediated actin assembly.

## Discussion

Here, we uncover a mechanism in response to confined migration-induced NE rupture, in which ATR-mediated formin activation and nuclear F-actin formation protect the nucleus from forces posed by the extracellular environment. Our data support a concept in which a dynamic actin nucleoskeleton is able to maintain the structural integrity of the nucleus. Our findings support a novel role for an actin structure within the nuclear compartment, adding an additional layer of how polymeric actin can contribute to cell

migratory functions such as cancer cell invasion in confined environments (Papalazarou and Machesky, 2021).

We demonstrate that NE rupture triggers DIAPH3 translocation to the nucleus (Fig. 2C). We also observed a perinuclear enrichment of DIAPH3 consistent with previous observations (Shao et al, 2015). Of note, recent reports support a mode of ATR activation through mechanical deformation of the nucleus, allowing for localization on the NE (Kumar et al, 2014). Thus, the perinuclear region might be an important sensory hub for DIAPH3 function.

ATR has been shown to phosphorylate a number of substrates in response to DNA damage (Matsuoka et al, 2009) and has been shown to promote nuclear actin polymerization upon osmotic and mechanical stress via recruitment of Filamin-A to the NE through binding with ATR-phosphorylated RASSF1A (Chatzifrangkeskou et al, 2025). Here, we demonstrate that in the events of confined migration-induced NE rupture, ATR mediates DIAPH3 phosphorylation at Ser1072 to promote nuclear actin polymerization, independently of the canonical ATR-DNA damage response pathway. However, whether DIAPH3 is indeed a direct substrate of ATR or whether ATR kinase activity indirectly regulates

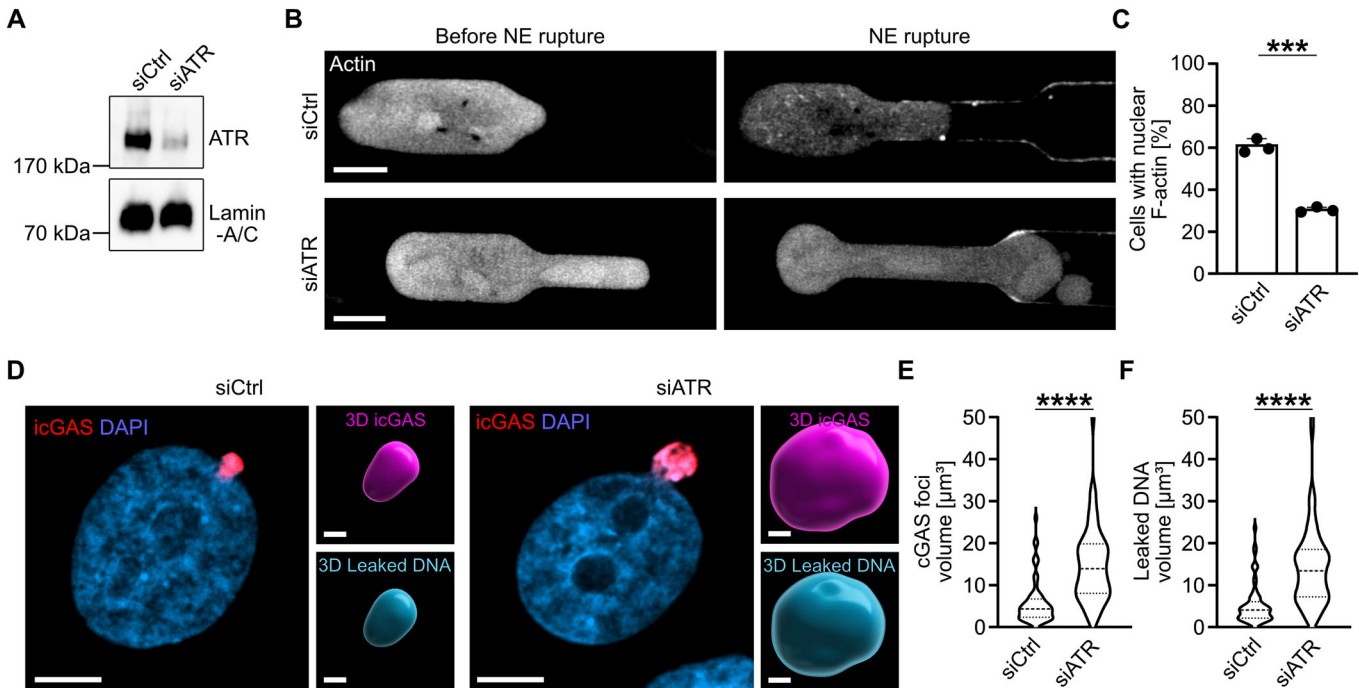

**Figure 5.    ATR is important for nuclear actin polymerization.**

(A) Representative immunoblots for the silencing efficiency of ATR (siATR). Lamin-A/C serves as a loading control. (B) Representative image sequences of HT1080-nAC-GFP cells migrating through 3- or 4 µm microchannels after transfecting siCtrl or siRNA against ATR. Before NE rupture; corresponds to the time before a NE rupture event had occurred. NE rupture; corresponds to the first frame in which a NE rupture event is detected via leakage of nAC-GFP. Actin is visualized via nAC-GFP. Scale bar, 5 µm. (C) Percentage of HT1080 cells stably expressing nAC-GFP that migrate through microchannels and display nuclear F-actin upon treatment with either control siRNA (siCtrl), or siRNA against ATR (siATR). Data shown as mean ± s.d. Three independent experiments per condition with $n \geq 10$ cells per experiment. Statistical analysis was performed with unpaired parametric $t$-test. Exact $P$ value: 0.00012. (D) Representative immunofluorescence images of HT1080 cells stably expressing icGAS-mCherry after being treated with either siCtrl or siATR. (Right) 3D rendering of icGAS-mCherry (magenta) and leaked chromatin (blue). Scale bar, 5 µm. Scale bar for 3D-rendered icGAS and 3D leaked DNA, 1 µm. (E) Quantification of 3D-rendered icGAS-GFP foci volume from (D). (F) Quantification of 3D-rendered DAPI volume in the cytoplasm from (D). Data from $n \geq 9$ cells were displayed in a violin plot with median and quartiles in three independent experiments. Statistical analysis was performed with Mann–Whitney test. Exact $P$ value from (E): $1.0 \times 10^{-12}$. Exact $P$ value from (F): $1.0 \times 10^{-12}$. Images are shown as maximum z-projections. Scale bar, 5 µm. ***$P < 0.001$, ****$P < 0.0001$. Source data are available online for this figure.

phosphorylation of DIAPH3 at S1072 requires further investigation.

Loss of ATR impacts nuclear stiffness, leading to softer nuclei (Kidiyoor et al, 2020). Consistent with this, we show here that ATR controls nuclear formin activity to steer nuclear rigidity via nuclear F-actin formation, providing an actin nucleoskeleton-based mechanism for regulating nuclear stiffness.

Here, using high spatiotemporal imaging we found that NE rupture precedes nuclear actin polymerization (Fig. 1G) as well as nuclear translocation of DIAPH3 (Fig. 2C). It is thus conceivable that on the one hand ATR is activated during nuclear deformation during confined migration, while on the other hand NE rupture promotes the entry of DIAPH3 serving as a substrate for ATR-mediated phosphorylation to promote nuclear actin assembly. Here, we were able to show that neither impairment of nuclear actin polymerization (Fig. 3F–I) nor ATR pharmacological inhibition (Fig. EV3B) had any impact on the frequency of NE rupture in migrating cancer cells. These observations support a model in which confined migration induced the NE rupture and subsequent ATR-mediated nuclear actin assembly sustains chromatin leakage for genome protection.

Previous studies have suggested that replication stress-mediated nuclear F-actin formation is sensitive to pharmacological inhibition of ATR as well as Arp-2/3 using VE-822 or CK-666 respectively (Lamm et al, 2020).

Using high spatiotemporal live-cell imaging and AFM, we found that formin-mediated nuclear actin filaments are required for the mechanical support of the nuclear organelle. In the case of replication-induced nuclear actin bundles (Lamm et al, 2020), the nuclear localization and dynamics of Arp-2/3 remain unresolved, while our data unambiguously demonstrate that Diaph3 rapidly accumulates in the nucleus in response to NE rupture to facilitate the formation of highly dynamic, distinct and transient nuclear actin structures.

Cells of the immune system, such as dendritic cells, also undergo NE rupture when presented with topologically complex environments to navigate (Alraies et al, 2024; Raab et al, 2016). These cells have developed mechanisms to identify paths of the least resistance (Renkawitz et al, 2019) and a perinuclear actin network to protect the structure of the nucleus during migration through tight pores (Thiam et al, 2016). Thus, further studies will be needed to investigate whether nuclear actin assembly is an additional mechanism to promote confined migration of immune cells and

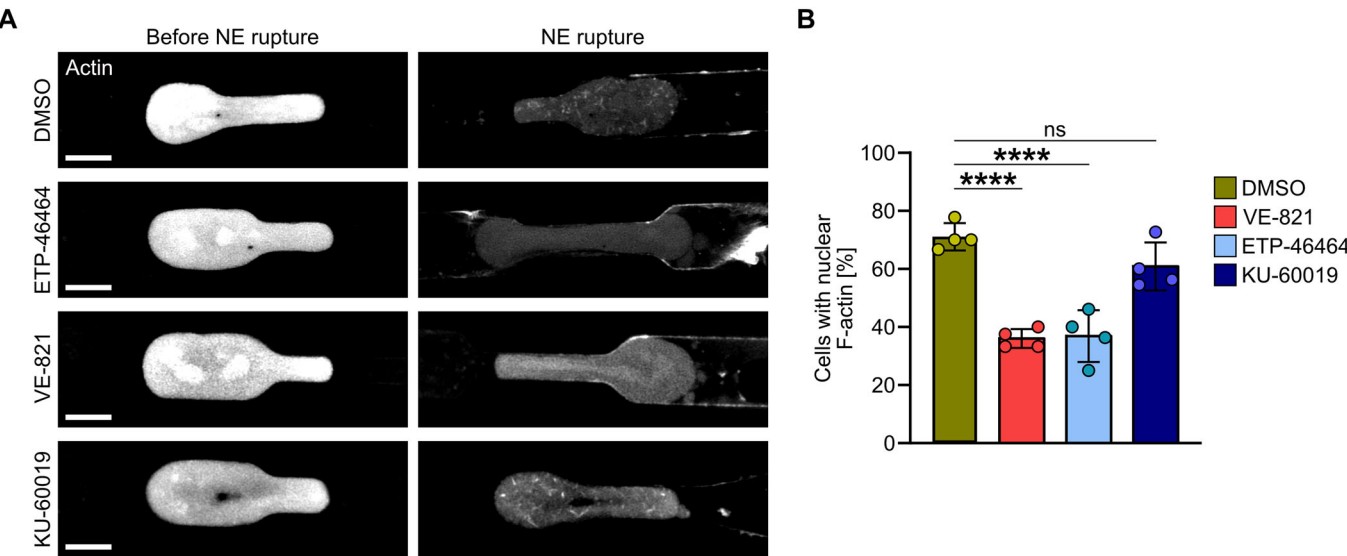

**Figure 6.  ATR regulates nuclear actin polymerization upon NER.**

(**A**) Image sequences of representative HT1080-nAC-GFP cells after treatment with either 0.01% DMSO, 3 µM ETP-46464, 0.1 µM VE-821, or 10 µM KU-60019. Before NE rupture; corresponds to the time before a NE rupture event had occurred. NE rupture; corresponds to the first frame in which a NE rupture event is detected via leakage of nAC-GFP. Actin is visualized via nAC-GFP. Scale bar, 5 µm. (**B**) Percentage of HT1080 cells stably expressing nAC-GFP migrating through microchannels displaying nuclear F-actin upon treatment with either DMSO (0.01%), ATR inhibitors (3 µM ETP-46464, 0.1 µM VE-821) or ATM inhibitor (10 µM KU-60019). Data shown as mean ± s.d. Four independent experiments per condition with $n \geq 8$ cells per experiment. Statistical analysis was performed by one-way analysis of variance (ANOVA). Exact $P$ values from left to right: $P = 0.00002$. $P = 0.00003$. $P = 0.123982$. ****$P < 0.0001$ and ns not significant. Source data are available online for this figure.

whether the lack of this protective mechanism could contribute to immune deficiencies.

In summary, our data support a model in which mechanical activation of ATR under confinement and subsequent NE rupture, leading to formin nuclear entry and phosphorylation promotes the formation of an actin nucleoskeleton that acts as a structural component of the nucleus to counteract organelle damage. Thus, our findings uncover a dynamic actin nucleoskeleton that provides mechanical support and nuclear stiffness in cells encountering physical obstacles that impose extensive deformations on the nucleus. Further studies will be needed to investigate the roles of the actin nucleoskeleton in other diseases that display altered NE properties and subsequent NE rupture, including laminopathies (De vos et al, 2011; Earle et al, 2020).

## Methods

### Reagents and tools table

| Reagent/resource | Reference or source | Identifier or catalog number |
|---|---|---|
| **Experimental models** | | |
| HT1080-WT | ATCC | CCL-121 |
| HT1080-nAC-GFP | Grosse lab | N/A |
| HT1080-icGAS-mCherry | This study | N/A |
| HT1080-icGAS-GFP/H2B-mCherry | This study | N/A |
| HT1080-icGAS-GFP/H2B-mCherry/NLS-BFP | This study | N/A |
| HT1080-icGAS-GFP/H2B-mCherry/NLS-BFP-Actin-R62D | This study | N/A |

| Reagent/resource | Reference or source | Identifier or catalog number |
|---|---|---|
| HEK293T-WT | ATCC | CRL-3216 |
| **Recombinant DNA** | | |
| pWPXL-nAC-GFP | Plessner et al, 2015 (https://doi.org/10.1074/jbc.M114.627166) | N/A |
| pCDH-icGAS-mCherry | Denais et al, 2016 (https://doi.org/10.1126/science.aad7297) | Addgene #132771 |
| pWPXL-icGAS-GFP | This study | N/A |
| pcDNA-GFP-Diaph3 | This study | N/A |
| pWPXL-H2B-mCherry | Baarlink et al, 2017 (https://doi.org/10.1038/ncb3641) | N/A |
| pEF-myc-NLS-BFP-Diaph3-WT | This study | N/A |
| pEF-myc-NLS-BFP-Diaph3-S1072D | This study | N/A |
| pEF-myc-NLS-BFP-Diaph3-S1072A | This study | N/A |
| pEF-NLS-mScarlet-actin-WT | This study | N/A |
| pEF-NLS-mScarlet-actin-R62D | This study | N/A |
| pEF-NLS-mScarlet-actin-S14C | This study | N/A |
| pEF-NLS-mScarlet-actin | This study | N/A |
| pEF-myc-NLS-mScarlet-Diaph3-WT | This study | N/A |
| pEF-myc-NLS-mScarlet-Diaph3-S1072D | This study | N/A |
| pEF-myc-NLS-mScarlet-Diaph3-S1072A | This study | N/A |
| pCMV-DN.KASH-mScarlet | | |
| pCMV-mScarlet | This study | N/A |
| pEF-NLS-BFP | This study | N/A |

| Reagent/resource | Reference or source | Identifier or catalog number |
|---|---|---|
| pWPXL-FMNL2-GFP | Wang et al, 2015 (https://doi.org/10.1016/j.devcel.2015.06.015) | N/A |
| pEF-Flag-NLS-Actin-WT | Baarlink et al, 2017 (https://doi.org/10.1038/ncb3641) | N/A |
| pEF-Flag-NLS-Actin-R62D | Baarlink et al, 2017 (https://doi.org/10.1038/ncb3641) | N/A |
| **Antibodies** | | |
| DIAPH1 | Abcam | ab129167 |
| DIAPH2 | Sigma | HPA005647 |
| DIAPH3 | Proteintech | 14342-1-AP |
| INF2 | Proteintech | 20466-1-AP |
| Tubulin | Cell Signaling Technology | 2125S |
| Lamin-A/C | Cell Signaling Technology | 4777S |
| Lamin B1 | Abcam | Ab16048 |
| ATR | NovusBio | NB100-308 |
| Phospho-Chk1 | Cell Signaling Technology | 2360S |
| Total-Chk1 | Cell Signaling Technology | 2360S |
| Phosphoserine | Abcam | Ab9332 |
| Myc | Cell Signaling Technology | 2276S |
| FLAG-M2 | Sigma | F1804 |
| Anti-rabbit IgG HRP | Cell Signaling Technology | No. 7074 |
| Anti-mouse IgG HRP | Cell Signaling Technology | No. 7076 |
| **Oligonucleotides and other sequence-based reagents** | | |
| siRNA against DIAPH1 - CAGCCGCTGCTGGATGGATTA | Qiagen | SI04345880 |
| siRNA against DIAPH2 - ACCGTCGAAAGCGGATTCCAA | Qiagen | SI00091329 |
| siRNA against DIAPH3 - CTCCGGCACAATTCAGTTCAA | Qiagen | SI04345880 |
| siRNA against INF2 - CGGCAAGATGTCGGTGAAGGA | Qiagen | SI04957057 |
| ON-TARGETplus siRNA SMARTpool against ATR | Dharmacon | L-003202-00-0005 |
| Forward PCR primer pWPXL-icGAS-GFP | This study | atatatacgcgtatgcagccttggcacggaaag |
| Reverse PCR primer pWPXL-icGAS-GFP | This study | tatataactagtttacttgtacagctcgtccatgcc |
| Forward PCR primer pWPXL-NLS-BFP | This study | atatatacgcgtatgccggagcagaagctgatatc |
| Reverse PCR primer pWPXL-NLS-BFP | This study | agctatactagttcaaagcttgtgcccc |
| Forward PCR primer pWPXL-NLS-BFP-actin-R62D | This study | atatatacgcgtatgccggagcagaagctgatatcc |
| Reverse PCR primer pWPXL-NLS-BFP-actin-R62D | This study | tatataactagtctagaagcatttgcggtggacgatg |
| Quickchange PCR for pEF-NLS-BFP-Diaph3-S1072D | This study | cttttgagaacaggcctctggtccatcgggctgagactctgccgaatatc |
| Quickchange PCR for pEF-NLS-BFP-Diaph3-S1072A | This study | cttttgagaacaggcctctgagccatcgggctgagactctgccgaatatc |
| PCR primers for Gibson cloning of NLS-mScarlet-actin-R62D – mScarlet.forward | This study | gaaggtggggatgagctgagcaagggcggaggca |
| PCR primers for Gibson cloning of NLS-mScarlet-actin-R62D – mScarlet.reverse | This study | cgtacctgctcgacatgttcaggcctgagtctagag |

| Reagent/resource | Reference or source | Identifier or catalog number |
|---|---|---|
| PCR primers for Gibson cloning of NLS-mScarlet-actin-R62D – pEF.forward | This study | tccggactcagatctcgagctatgg |
| PCR primers for Gibson cloning of NLS-mScarlet-actin-R62D – pEF.reverse | This study | gctcatccccaccttccgct |
| Quickchange PCR for pmScarlet-NLS-mScarlet-actin-WT | This study | ccagagcaagagaggcatcctcacc |
| Quickchange PCR for pmScarlet-NLS-mScarlet-actin-S14C | This study | tcgacaacggctgcggcatgtgca |
| PCR primers for Gibson cloning of pEF-myc-NLS-mScarlet-Diaph3-WT – mScarlet.forward | This study | gcaacctcaaacagacaccatgccggagcagaagctgatatccg |
| PCR primers for Gibson cloning of pEF-myc-NLS-mScarlet-Diaph3-WT – mScarlet.reverse | This study | gaattcggatccggccatcttgtacagctcgtccatgccg |
| PCR primers for Gibson cloning of pEF-myc-NLS-mScarlet-Diaph3-WT – pEF.forward | This study | atggccggatccgaattcgtc |
| PCR primers for Gibson cloning of pEF-myc-NLS-mScarlet-Diaph3-WT – pEF.reverse | This study | ggtgtctgtttgaggttgctagtgaacac |
| PCR primers for Gibson cloning of pcDNA-GFP-Diaph3 – pCDNA.reverse | This study | ggtggcggatccgagctc |
| PCR primers for Gibson cloning of pcDNA-GFP-Diaph3 – pCDNA.forward | This study | taaacccgctgatcagcctcgac |
| PCR primers for Gibson cloning of pcDNA-GFP-Diaph3 – Diaph3.forward | This study | ggcatggacgagctgtacaagaatccgcctaagaaaaagcggaaggtgtc |
| PCR primers for Gibson cloning of pcDNA-GFP-Diaph3 – Diaph3.reverse | This study | gctttttcttaggcggattcttgtacagctcgtccatgccgag |
| PCR primers for Gibson cloning of pcDNA-GFP-Diaph3 – GFP.forward | This study | gagctcggatccgccaccatggtgagcaagggcgaggagc |
| PCR primers for Gibson cloning of pcDNA-GFP-Diaph3 – GFP.reverse | This study | gcttttcttaggcggattcttgtacagctcgtccatgccgag |
| **Chemicals, Enzymes and other reagents** | | |
| Lipofectamine 3000 | Invitrogen | L3000001 |
| Lipofectamine RNAiMAX | Invitrogen | 13778075 |
| Q5 High-Fidelity DNA Polymerase | New England Biolabs | M0491S |
| T4 DNA ligase | New England Biolabs | M0202T |
| Bovine serum albumin | Roth | 1ET5.3 |
| DMEM High Glucose w/ stable glutamine w/sodium pyruvate | Anprotec | AC-LM-0013 |
| Penicillin/Streptomycin | Anprotec | AC-AB-0024 |
| Fetal calf serum | Anprotec | AC-SM-0190 |
| Bovine fibronectin | Sigma-Aldrich | F1141 |
| MluI-HF | New England Biolabs | R3198L |
| SpeI-HF | New England Biolabs | R3133L |
| cOmplete™, EDTA-free Protease Inhibitor Cocktail | Roche | 11873580001 |
| Power blotter 1-Step™ transfer buffer (5X) | Invitrogen | PB7300X3 |
| SuperSignal West Femto maximum Sensitivity substrate | Thermo Fisher Scientific | 34096 |
| SuperSignal West Dura Extended Duration Substrate | Thermo Fisher Scientific | 34075 |
| Transfer membrane ROTI®PVDF | Roth | T830.1 |
| ProLong Diamond Antifade Mountant | Invitrogen | P36961 |
| DAPI | Sigma-Aldrich | D9542 |
| VenorGem Classic Mycoplasma Detection Kit | Minerva Biolabs | 11-1100 |

| Reagent/resource | Reference or source | Identifier or catalog number |
|---|---|---|
| CK-666 | Sigma | SML006-5MG |
| Y27132 | Tocris | Cat. No. #1254 |
| ETP-46464 | Selleck Chem | S8050 |
| VE-821 | Selleck Chem | S8007 |
| KU-60019 | Abcam | Ab144817 |
| LY2603618 | Merck | SML2855 |
| SPY650-FastActX | TebuBio | |
| Anhydrous DMSO | Invitrogen | D12345 |
| DpnI | New England Biolabs | R0176L |
| PhosSTOP™ | Roche | 4906845001 |
| Myc-agarose beads | Santa Cruz | sc-500771 |
| Triangular non-conductive silicon nitride cantilevers | Bruker | MLCT-E |
| 35 mm cell culture dishes | TPP | 93040 |
| Microchannel dishes with constrictions | 4D-Cell | MC019 |
| Sera-Mag SpeedBead magnetic carboxylate-modified particles | Cytiva | |
| AttractSPE® Disks Bio RPS | AFFINISEP | SPE-Disks-Bio RPS-M-47.20 |
| **Software** | | |
| GraphPad Prism 10.1.0 | https://www.graphpad.com/ | |
| ZenBlue | Zeiss | |
| ImageJ/Fiji | https://imagej.net/software/fiji/ | |
| Imaris 10.1.0 | Oxford Instruments | |
| Snapgene | https://www.snapgene.com/ | |
| MaxQuant version 2.0.2.0 | https://www.maxquant.org/ | |
| JPK software Version 8.0.144 | JPK | |
| **Other** | | |
| LSM800 confocal laser-scanning microscope | Zeiss | |
| ELYRA 7 microscope | Zeiss | |
| NanoWizard 4 XP AFM system | Bruker | |
| Zeiss Axiovert 200 M fluorescence microscope | Zeiss | |
| UltiMate™ 3000 RSLCnano coupled to Q Exactive mass spectrometer | Thermo Fisher Scientific | |
| PepMap C18 trapping column | Thermo Fisher Scientific | |
| µPAC™ C18 pillar array column | Thermo Fisher Scientific | |
| Nanospray Flex ion source with a liquid Junction™ PST-HV-NFU and a fused silica emitter | Thermo Fisher Scientific | |
| Power blotter XL system | Invitrogen | PB0013 |

## Cell culture, transfection, and RNA interference

Human HT1080 cells (ATCC) and HEK293T (ATCC) as well as derivative cell lines were maintained in Dulbecco´s modified Eagle´s medium (DMEM, anprotec) supplemented with 10% fetal calf serum (FCS, anprotec) and 100 units ml$^{-1}$ penicillin/streptomycin (anprotec) at 37 °C and 5% $CO_2$ atmosphere. All cell lines were tested for mycoplasma contamination on a monthly basis using the

VenorGem Classic Mycoplasma detection kit (Minerva Biolabs). Plasmid DNA transfections were performed using Lipofectamine 3000 (Invitrogen), according to the instructions of the manufacturer. For protein depletion via siRNA, cells were transfected with targeting or non-targeting siRNA using Lipofectamine RNAiMAX (Invitrogen). Experiments in which DIAPH1 and DIAPH3 or ATR had been depleted, were performed 48 h post-transfection. The following siRNAs were used:

*Ctrl*, All Stars Negative Control siRNA, AATTCTCCGAACGTGTCACGT (Qiagen, no. SI00073941);

*DIAPH1*, CAGCCGCTGCTGGATGGATTA (Qiagen, SI04345880)
*DIAPH2*, ACCGTCGAAAGCGGATTCCAA (Qiagen, SI00091329)
*DIAPH3*, CTCCGGCACAATTCAGTTCAA (Qiagen, SI04345880)
*INF2*, CGGCAAGATGTCGGTGAAGGA (Qiagen, SI04957057)
*ATR*, ON-TARGETplus siRNA SMARTpool (Dharmacon, L-003202-00-0005)

## Plasmids and reagents

For the generation of pWPXL-inactive-cGAS E225A/D227A-GFP (pWPXL-icGAS-GFP), the cDNA of icGAS-GFP was subcloned into a pWPXL backbone via insertion of MluI/SpeI restriction sites using the following primers (5'→3'): MluI_cGAS-GFP.fwd, atatatacgcgtatgcagccttggcacggaaag, and SpeI_cGAS-GFP.rev, tatataactagtttacttgtacagctcgtccatgcc. The insert was then ligated into an MluI/SpeI-digested pWPXL vector. pCDH-cGAS-E225A D227A-mCherry2-EF1-Puro (icGAS-mCherry) was a gift from Jan Lammerding (Addgene plasmid # 132771; http://n2t.net/addgene:132771; RRID: Addgene_132771). The pWPXL-NLS-BFP and pWPXL-NLS-BFP-actin-R62D (20) were generated following standard restriction enzyme cloning. NLS-BFP or NLS-BFP-actin-R62D were amplified with the insertion of SpeI/MluI sites using the following primers: MluI_NLS-BFP.fwd, atatatacgcgtatgccggagcagaagctgatatc, SpeI_NLS-BFP.rev, agctatactagttcaaagcttgtgcccc, MluI_NLS-R62D.fwd, atatatacgcgtatgccggagcagaagctgatatcc, SpeI_NLS-R62D.rev, tatataactagtctctagaagcatttgcggtggacgatg. The insert was then ligated into an MluI/SpeI-digested pWPXL vector. pEF-NLS-BFP-Diaph3 was generated as described previously (20). pEF-NLS-BFP-Diaph3-S1072D and pEF-NLS-BFP-Diaph3-S1072A were generated by QuickChange PCR using the following primers: S1072D.rev, ctttgagaacaggcctctggtccatcgggctgagactctgccgaatatc, and S1072A.rev, ctttgagaacaggcctctgagccatcgggctgagactctgccgaatatc. Flag-NLS-Actin-WT, Flag-NLS-Actin-R62D, pWPXL-H2B-mCherry (21) and pWPXL-nAC-GFP were generated as described previously (17).

pEF-NLS-mScarlet-actin-R62D was subcloned from pEF-NLS-BFP-actin-R62D and pmScarlet-DAAM2-wt (18) using the Gibson cloning technique (NEB). The following primers (5' → 3') were used: mScarlet fwd: gaaggtggggatgagcgtgagcaagggcgaggca; mScarlet rew: ccgtacctgctcgacatgttcaggcctgagtctagag; backbone fwd: tccggactcagatctcgagctatgg; backbone rev: gctcatccccaccttccgct.

pEF-NLS-mScarlet-actin was subcloned from pEF-NLS-mScarlet-actin-R62D with QuickChange PCR. The following primer (5' → 3') was used: QC R62D to R62: ccagagcaagagaggcatcctcacc. pEF-NLS-mScarlet-actin-S14C and NLS-mScarlet were subcloned from NLS-mScarlet-actin with QuickChange PCR. The following primers (5' → 3') were used: QC S14C: tcgacaacggctgcggcatgtgca; QC pEF-NLS-mScarlet: catggacgagctgtacaagtaatccggactcagatctcgagctatggatg. pEF-NLS-mScarlet-Diaph3-wt/mDia2-S1072A and

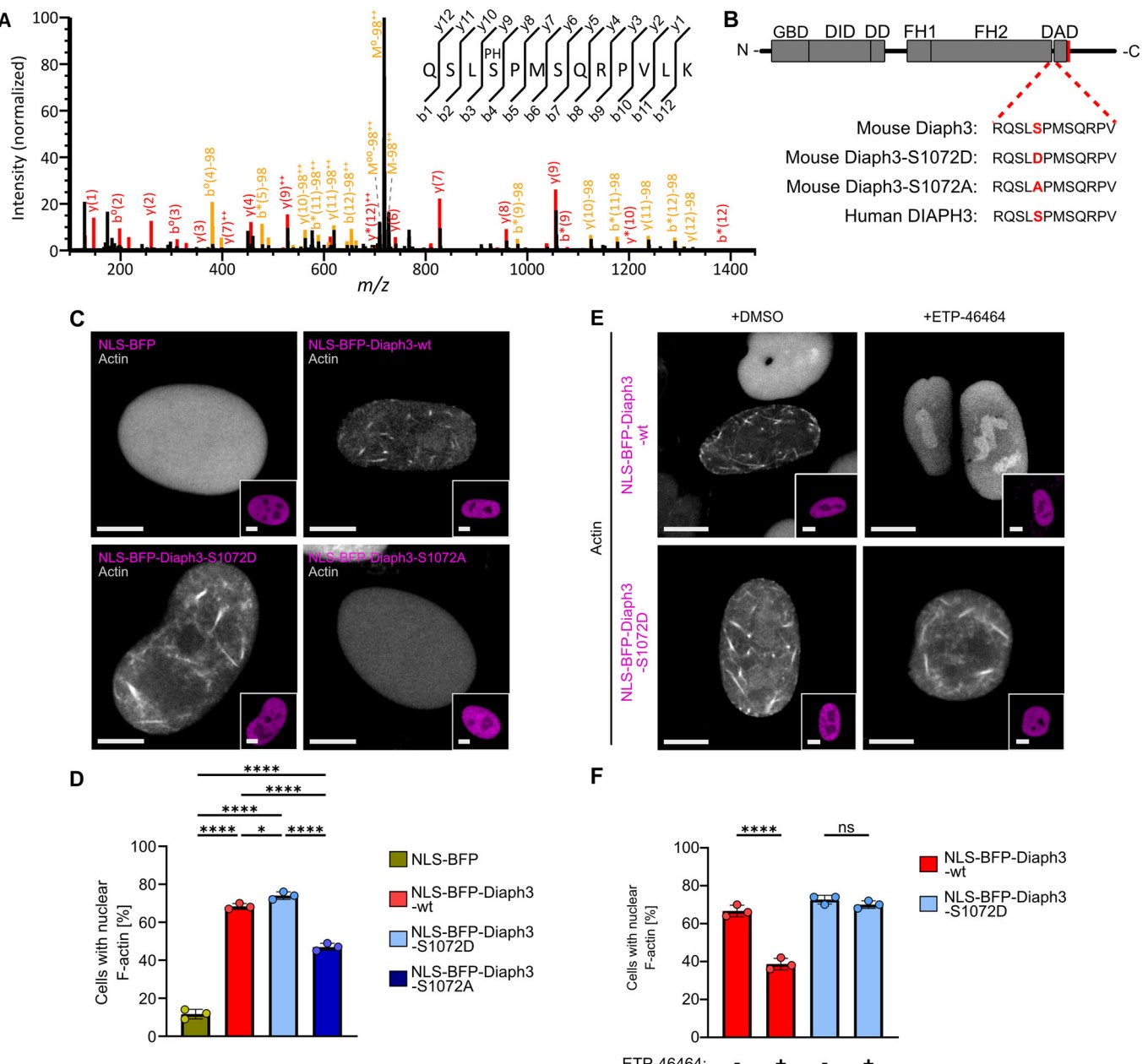

© The Author(s)

Diaph3-S1072D were subcloned from pEF-NLS-BFP-Diaph3-wt/ Diaph3-S1072A and mDiaS1072D and pmScarlet-DAAM2-wt (18) using the Gibson cloning technique (NEB). The following primers (5' → 3') were used: mScarlet fwd: gcaacctcaaacagacaccatgccggag- cagaagctgatatccg; mScarlet rew: gaattcggatccggccatcttgtacagctcgtc- catgccg; backbone fwd: atggccggatccgaattcgtc; backbone rev: ggtgtctgtttgaggttgctagtgaacac. pcDNA-GFP-Diaph3 was subcloned from pEF-NLS-BFP-Diaph3-wt using the Gibson cloning technique (NEB). The following primers (5' → 3') were used: pcDNA.fowards: taaacccgctgatcagcctcgac; pcDNA.reverse: ggtggcggatccgagctc; GFP.forward: gagctcggatccgccaccatggtgagcaagggcgaggagc; GFP.re- verse: gctttttcttaggcggattcttgtacagctcgtccatgccgag; Diaph3.forward: ggcatggacgagctgtacaagaatccgcctaagaaaaagcggaaggtgtc; Diaph3.re- verse: gctttttcttaggcggattcttgtacagctcgtccatgccgag.

All plasmids have been subjected to verification by sequencing.

## Lentiviral transduction

Packaging plasmids pMD2.G and pSPAX, along with the desired lentiviral backbone, were transfected into HEK293T cells using Lipofectamine 3000 (Invitrogen) according to the manufacturer's instructions. The following day, the supernatant from the HEK293T cells, that contains the viral particles, was collected and applied to HT1080 cells that have been seeded in six-well dishes and were let to grow. Afterwards, the stable cell lines were FACS sorted to produce populations with homogenous expression levels.

## Immunoblotting

Cells were subjected to lysis directly in 1x Laemmli buffer and were heated at 95 °C for 10 min. Separation by sodium dodecyl sulfate

**Figure 7.  ATR regulates Diaph3 phosphorylation to drive nuclear actin polymerization.**

(A) MS/MS spectrum identifying the tryptic peptide Q1069-K1081 of Diaph3 carrying a phosphorylation (PH) at serine S1072. Fragments matched to masses of b and y ions are marked in the scheme and by red labels in the spectrum. Yellow labels denote matches for fragment or precursor (M) ions after neutral loss of the phosphate group. Neutral loss of $H_2O$ or NH is indicated by subscripts of ° or *. (B) Schematic representation of DIAPH3 functional domains and the identified consensus sequence, a S-Q motif (red) adjacent to DAD (Diaphanous-autoregulatory domain). The conserved sequence of 12 aminoacids displays the serine residue where ATR can phosphorylate human DIAPH3 or the mouse orthologue mDia2 (red). (C) Representative images of HT1080-nAC-GFP (white) that transiently express either NLS-BFP, NLS-BFP-Diaph3-wt, NLS-BFP-Diaph3-S1072D or NLS-BFP-Diaph3-S1072A with bottom right images showing nuclear expression of these constructs (magenta); Scale bar, 5 μm. (D) Percentage of HT1080 cells stably expressing nAC-GFP displaying nuclear F-actin formation after co-expressing NLS-BFP, NLS-BFP-mDia2-wt, NLS-BFP-Diaph3-S1072D, or NLS-BFP-Diaph3-S1072A. Data shown as mean ± s.d. Three independent experiments per condition with $n \geq 100$ cells per experiment. Statistical analysis was performed by one-way analysis of variance (ANOVA) with Tukey's multiple comparison. Exact P values for NLS-BFP vs NLS-BFP-Diaph3-wt, NLS-BFP-Diaph3-S1072D, and NLS-BFP-Diaph3-S1072A, respectively: $P = 5.0 \times 10^{-9}$. $P = 3.0 \times 10^{-9}$. $P = 9.0 \times 10^{-8}$. Exact P values for NLS-BFP-Diaph3-WT vs NLS-BFP-Diaph3-S1072D and NLS-BFP-Diaph3-S1072A are respectively: $P = 0.0371$. $P = 6.0 \times 10^{-6}$. Exact P values for NLS-BFP-Diaph3-S1072D vs NLS-BFP-Diaph3-S1072A: $P = 1.0 \times 10^{-6}$. (E) Representative images of HT1080-nAC-GFP (white) that transiently express either NLS-BFP-Diaph3-wt or NLS-BFP-Diaph3-S1072D in the presence or absence of ETP-46464, with bottom right images showing nuclear expression of these constructs (magenta); Scale bar, 5 μm. (F) Percentage of HT1080-nAC-GFP cells transiently expressing either NLS-Diaph3-wt or NLS-Diaph3-S1072D that display nuclear F-actin formation after being treated with either 0.01% DMSO or 3 μM ETP-46464 for 3 h. Data shown as mean ± s.d. Three independent experiments per condition with $n = 50$ cells per experiment. Statistical analysis between DMSO and ETP-46464 treatment after expression of NLS-BFP-Diaph3-WT was performed by an unpaired parametric t-test. Exact P value: $3.0 \times 10^{-7}$. Statistical analysis between DMSO and ETP-46464 treatment after expression of NLS-BFP-Diaph3-S1072D was performed by the Mann–Whitney test. Exact P value: 0.3000. ns: $P > 0.05$, $**P < 0.01$, $*** P < 0.001$, $**** P < 0.0001$. Source data are available online for this figure.

polyacrylamide gel electrophoresis (SDS-PAGE) was then applied after the lysates were equally loaded on separating gels of varying concentrations, based on protein size. Semi-dry transfer (Power Blotter XL System, Invitrogen) was performed according to the manufacturer's instructions. Afterwards, the PVDF membrane was blocked with 5% milk or 5% BSA for 1 h at room temperature under constant rocking and with the primary antibodies overnight at 4 °C. Horseradish peroxidase conjugated secondary antibody was applied for 1 h at room temperature, and protein detection was performed by using enhanced chemiluminescence (ECL) reagent (SuperSignal West Femto Maximum Sensitivity Substrate, Thermo Fisher Scientific). For quantification of immunoblots, SuperSignal West Dura Extended Duration Substrate was used for protein detection. Primary antibodies used were: mAB anti-DIAPH1 (1:1000, Abcam, ab129167), pAB anti-DIAPH2 (1:1000, Sigma, HPA005647), pAB anti-DIAPH3 (1:1000, Proteintech, 14342-1-AP), pAB anti-INF2 (1:1000, Proteintech, 20466-1-AP), mAB anti-Tubulin (1:1000, CST, 2125S), mAB anti-Lamin-A/C (1:1000, CST, 4777S), pAB anti-Lamin B1 (1:1000, Abcam, ab16048), mAB anti-ATR (1:2000, NovusBio, NB100-308), anti-rabbit mAB anti-pChk1 (1:1000, CST, 2348 T), anti-mouse mAB total-Chk1 (1:1000, CST, 2360S), rabbit anti-phosphoserine (1:2000, Abcam, ab9332), mouse anti-myc (1:1000, CST, 2276S), and anti-mouse mAB anti-FLAG-M2 (1:500, Sigma, F1804). Secondary antibodies used were: anti-rabbit IgG HRP (1:3000, CST, no. 7074) and anti-mouse IgG HRP (1:3000, CST, no. 7076).

## Live-cell imaging

ELYRA 7 microscope (Zeiss) equipped with a 63 × 1.4 Oil DIC objective and a Pecon incubation chamber was used for Structured illumination microscopy (SIM) imaging. Images were subjected to processing with ZEN 3.0 black edition (Zeiss). Wiener filter strength was automatically selected by the "standard" end criterion of the manufacturer.

## Microchannels loading and live-cell confocal microscopy

Firstly, microchannel dishes (4D-Cell, MC019) were rinsed with distilled water containing 1% P/S and subsequently coated with 50 μg/μl fibronectin overnight at 4 °C. The following day, the dish was

rinsed with PBS and was incubated with Dulbecco´s modified Eagle´s medium (DMEM, anprotec) supplemented with 1% fetal calf serum (FCS, anprotec) and 100 units ml$^{-1}$ penicillin/streptomycin (anprotec) at 37 °C and 5% $CO_2$ atmosphere for at least 1 h (here, drugs were added when necessary). Following that step, HT1080 cells ($1 \times 10^5$) were seeded in the seeding wells and were left to adhere at 37 °C and 5% $CO_2$ for at least 1 h. Cells were let to migrate through 3 μm microchannels or 3- and 4 μm microchannels in the case of silencing or pharmacological inhibition of ATR/ATM (Figs 5 and 6A,B). Then, the dish was filled with 1% FCS Full DMEM medium and were transferred to the microscope for live-cell imaging at 37 °C and 5% $CO_2$ atmosphere. Images were acquired with an LSM800 confocal laser-scanning microscope (Zeiss) equipped with a 63X, 1.4 NA oil objective and Airyscan detector, along with the Zen blue software (Zeiss). Cells in the microchannels were imaged as single z-planes. Images were taken in intervals of 5 min for nuclear actin assembly observation. For the sequence of events (Fig. 1E), images were acquired every 12 s. In the case of H2B-mCherry imaging for nuclear fragmentation and NE rupture frequency (Fig. 3A–H), images were acquired in 10 min intervals.

## Drug treatments

About 100 μM CK-666 (100 μM, Sigma, SML006-5MG), 10 μM Y27132 (10 μM, Tocris, Cat. No. #1254), ETP-46464(3 μM, Selleck Chem, S8050), VE-821(0.1 μM, Selleck Chem, S8007), KU-60019 (10 μM, Abcam, ab144817), LY2603618 (1 μM, Merck, SML2855), and SPY650-FastActX (1:1000, TebuBio). All drugs were dissolved in DMSO, and DMSO was used as a control for all of the drug treatments. If needed for microchannel experiments, drugs were further diluted in DMEM (Gibco) supplemented with 1% fetal calf serum (FCS) and 100 units ml$^{-1}$ penicillin/streptomycin, and microchannels were incubated for at least 1 h before cell loading. Cells were resuspended in medium containing the desired drug before being plated in the seeding wells.

## Quantification

To assist visualization of nuclear actin filaments upon NE rupture for quantification purposes, we applied background subtraction,

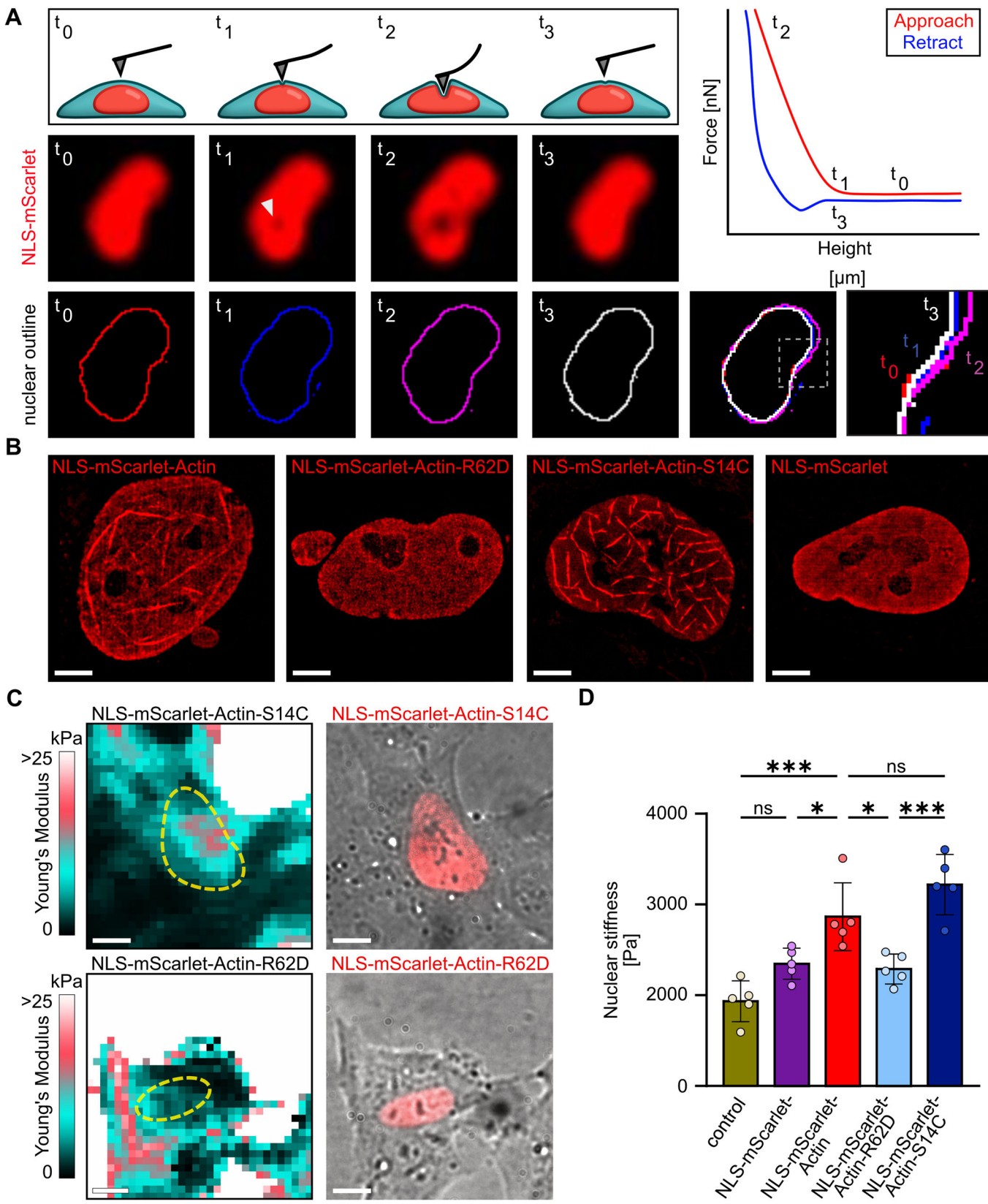

◄ **Figure 8. Nuclear actin polymerization regulates nuclear stiffness.**

(A) Schematic representation of probing the nucleus of live cells at four different time points (t$_0$-t$_3$, upper panel). The middle panel displays a nucleus expressing NLS-mScarlet during probing with the AFM. Bottom panel depicts the nuclear outline at corresponding time points. The merging of the nuclear outlines shows stretching of the nucleus during the experiment. The right graph represents a typical force-distance curve acquired from a force spectroscopy experiment. t$_0$ = initial position of AFM cantilever and subsequent approach. t$_1$ = first contact between the cantilever tip and the nucleus of the cell. Arrow in NLS-mScarlet panel indicates indentation point of cantilever tip. t$_2$ = point of maximum force and at the same time the deepest indentation of the nucleus. t$_3$ = retract and the end of the force spectroscopy measurement. (B) Nuclei of living cells transiently expressing NLS-mScarlet-Actin, the non-polymerizable NLS-mScarlet-Actin-R62D, the F-actin forming NLS-mScarlet-Actin-S14C or NLS-mScarlet, acquired with structured illumination microscopy (SIM) imaging. Cells expressing NLS-mScarlet-Actin and NLS-mScarlet-Actin-S14C display actin filaments, and cells expressing NLS-mScarlet-Actin-R62D and NLS-mScarlet show no actin filaments or filamentous structures. Scale bar, 2 μm. (C) Force map of a cell transiently expressing NLS-mScarlet-Actin-S14C (top panel) and a cell expressing NLS-mScarlet-Actin-R62D (bottom panel). The dashed yellow circle represents the position of the nucleus in the force map. Young's moduli acquired from the force-maps indicate a stiffer nucleus in NLS-mScarlet-Actin-S14C expressing cells. Scale bar, 10 μm. (D) Young's moduli from nuclei of cells transiently expressing NLS-mScarlet, NLS-mScarlet-tagged Actin, -Actin-S14C or -Actin-R62D. Control cells were mock-transfected. Cells expressing NLS-mScarlet-Actin and NLS-mScarlet-Actin-S14C exhibit significantly stiffer nuclei in comparison to NLS-mScarlet-Actin-R62D, NLS-mScarlet and control cells. n = 5. Data shown as mean ± s.d. Each dot represents the average Young's modulus of at least 30 nuclei. Statistical analysis was performed by one-way analysis of variance (ANOVA) with Tukey's multiple comparison. Exact P values from left to right: P = 0.0002. P = 0.0424. P = 0.0205. P = 0.0002. *P < 0.05, **P < 0.01, ***P < 0.001, and ns not significant. Source data are available online for this figure.

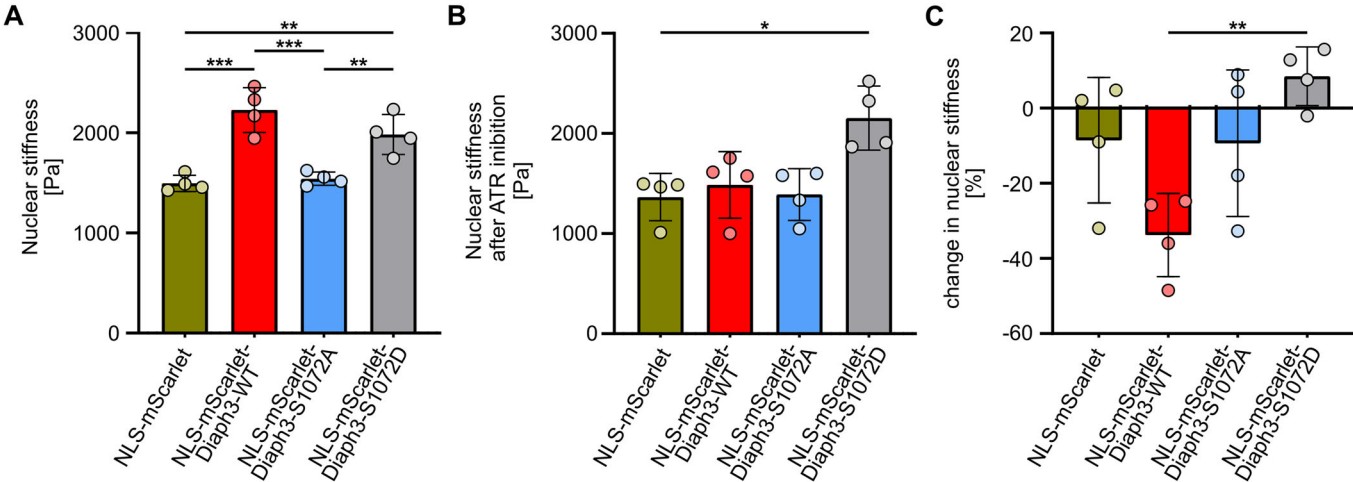

**Figure 9. ATR-mediated formin activation controls nuclear stiffness.**

(A) Nuclear stiffness of cells transiently expressing NLS-mScarlet, NLS-mScarlet-Diaph3-wt, NLS-mScarlet-Diaph3-S1072A and NLS-mScarlet-Diaph3-S1072D was measured with AFM. Data shown as mean ± s.d. Statistical analysis was performed by one-way analysis of variance (ANOVA) with Tukey's multiple comparison. Exact P value for NLS-mScarlet vs NLS-mScarlet-Diaph3-S1072D: P = 0.0045. Other P values from left to right: P = 0.00014. P = 0.0003. P = 0.0092. (B) Nuclear stiffness after ATR inhibitor ETP-46464 was added (3 μM, 3 h) in the same dish. Data shown as mean ± s.d. Statistical analysis was performed by the Kruskal–Wallis test. Exact P value: P = 0.0286. (C) Quantification of the change in Young's modulus (YM) after ATR inhibition. Nuclear stiffness of cells expressing NLS-mScarlet-Diaph3-S1072D and NLS-mScarlet-Diaph3-S1072A is unaffected by ETP-46464 treatment. Nuclei of cells expressing NLS-mScarlet-Diaph3-wt show a significant reduction in YM in comparison to NLS-Diaph3-S1072D and NLS-Diaph3-S1072A expressing cells. Data shown as mean ± s.d. Statistical analysis was performed by one-way analysis of variance (ANOVA) with Tukey's multiple comparison. Exact P values: P = 0.0068. *P < 0.05, **P < 0.01. ***P < 0.001. Source data are available online for this figure.

median and Gaussian filtering. Cells displaying nuclear F-actin were scored as positive. Cells that were negative for nuclear F-actin formation were scored as negative. To remove subjective bias, files were randomized by removing the file names prior to quantification. To quantify the frequency of cells displaying nuclear F-actin, the number of nuclear F-actin-positive cells were divided by the total number of cells observed. For the NE rupture frequency, the leakage of nAC-GFP from the nucleus to the cytoplasm was qualitatively assessed binarily (yes/no). The number of cells that underwent NE rupture were divided by the total number of cells observed, expressed as a percentage of all cells. For quantification of cells undergoing NE rupture as observed by accumulation of icGAS, cells that showed enrichment of icGAS in the perinuclear area were classified as ruptured cells. The cells that displayed no

enrichment were scored as negative. Then the number of icGAS-positive cells was divided by the total number of cells observed, and expressed as a percentage of cells. For the calculation of the time lag between NE rupture and nuclear F-actin formation, the time in which nuclear F-actin formation is observed was subtracted from the time in which NE rupture was observed by loss of nAC-GFP signal from the nucleus (t$_{lag}$ = t$_{nuc.F-actin}$ − t$_{NE\ rupture}$). To quantify the GFP-Diaph3 translocation to the nucleus in response to NE rupture, the mean fluorescence intensity of GFP-Diaph3 was measured after NE rupture and after Diaph3 export and normalized to the mean fluorescence intensity of GFP-Diaph3 before a NE rupture event occurred. To quantify the frequency with which nuclear fragmentation was observed after migration through 3 μm, cells that displayed a fragmented nucleus were scored as cells with

fragmented nuclei. Then this number was divided by the total number of cells that were observed, and expressed as a percentage.

## Immunofluorescence

HT1080 cells stably expressing icGAS-mCherry were grown in six-well plates that contained fibronectin-coated coverslips. Twenty-four hours later, the cells were fixed in 4% paraformaldehyde (PFA) for 10 min at 4 °C. The fixative agent was removed by washing with 1x PBS three times, and the cells were permeabilized with 0.02% Triton X-100 in PBS for 10 min at room temperature. After blocking with 5% BSA, the coverslips were incubated with the primary antibodies diluted in 5% BSA solution overnight at 4 °C in a humid chamber. The next day, the coverslips were incubated with fluorescently labeled secondary antibody along with DAPI (1:1000) for 1 h at room temperature in a humid chamber, before mounting the coverslips onto glass slides using ProLong Diamond Antifade Mountant (Invitrogen). Secondary antibody used was Alexa Fluor 488 goat anti-mouse IgG (H + L) (1:1000, Invitrogen, A11001). Fluorescence images were generated using an LSM880 confocal laser-scanning microscope (Zeiss) equipped with a 63x, 1.4 NA oil objective and an Airyscan detector, along with blue Zen software (Zeiss). Images were acquired as z-stacks with 0.22 μm intervals. All images acquired for each experiment were subjected to Airyscan processing using consistent Super Resolution parameters.

## icGAS-mCherry volume

For the quantification of the icGAS foci volume (Figs. 4A–G and 5D–G), HT1080-icGAS-mCherry cells were seeded in fibronectin-coated coverslips. Twenty-four hours later, the cells were treated with either siCtrl or siDIAPH1/3. Forty-eight hours later, the coverslips were fixated in 4% PFA for 10 min at room temperature. The fixative was washed with PBS, followed by permeabilization with 0.02% Triton X-100 in PBS for 10 min at room temperature and blocking in 5% BSA for 1 h at room temperature. Afterward, the coverslips were stained only with DAPI (1:1000) for 1 h, followed by mounting onto glass slides using ProLong Diamond Antifade Mountant (Invitrogen). Fluorescence images were generated using a LSM880 confocal laser-scanning microscope (Zeiss) equipped with a 63x, 1.4 NA oil objective and airyscan detector, along with blue Zen software (Zeiss). Images were acquired as z-stacks with 0.22 μm intervals. All images acquired for each experiment were subjected to Airyscan processing using constant super-resolution parameters. 3D rendering of icGAS-mCherry foci in siCtrl or siDIAPH1/3, siCtrl or siATR-treated cells and NLS-Actin-WT or NLS-Actin-R62D transfected cells was performed, and the nuclear volume of each focus was extracted. Leaked DNA volume was quantified using the icGAS 3D render as a mask. The voxel intensity of DAPI outside of the icGAS mask was set to 0, and the volume of DAPI inside the icGAS mask was quantified and termed as "leaked DNA volume". 3D rendering of icGAS-mCherry foci and DAPI was performed using the surface tool of IMARIS 10.1.0, and the volume was extracted.

## Immunoprecipitation

All steps were carried out on ice unless indicated otherwise, with centrifugation performed at 4 °C. HT1080-wt cells were seeded on six-well plates, transfected with myc-NLS-BFP-Diaph3-wt or myc-NLS-BFP-Diaph3-S1072A using Lipofectamine 3000 (Thermo Fisher Scientific) according to the manufacturer's instructions. The next day, they were treated with 10 mM HU for 3 h in the presence or absence of ETP-46464 (3 h treatment) and were harvested by scraping into PBS. After washing with PBS at 800x G, the cells were lysed in Ripa buffer (20 mM Tris, pH 7.5, 0.15 M NaCl, 1 mM Na$_2$EDTA, 1 mM EGTA, 1% NP-40, 2.5 mM Na-pyrophosphate, 1 mM Na$_3$VO$_4$, 1 mM β-glycerophosphate, 1 μg/mL leupeptin—with 1% dexocholate, 1× cOmplete protease inhibitor cocktail EDTA-free (Roche) and PhosSTOP™ (Sigma). The suspension was left on ice for 20 min, vortexing every 5 min. The lysates were incubated with myc-coated agarose beads (Santa Cruz) for 2 h while rotating at 4 °C. The samples were then washed three times with 1x Ripa containing 1% NP-40, and after washing, they were centrifuged at $1000 \times g$ for 3 min.

## Sample preparation for identification of phosphorylation sites by LC-MS/MS

Eluted proteins were prepared for nanoLC-MS/MS analysis using SP3 on-bead digestion with trypsin according to Hughes et al (2019). Cysteine residues were reduced with 10 mM DTT for 30 min at 60 °C and alkylated with 20 mM chloroacetamide for 20 min at 37 °C. Proteins were absorbed to the beads (Sera-Mag SpeedBead magnetic carboxylate-modified particles, Cytiva) with a 1:10 final protein:bead (w:w) ratio in 80% ethanol for 5 min at 24 °C and 1000 rpm. Digestion was performed in 50 μl of 100 mM ammonium bicarbonate containing 0.1 μg trypsin (Promega V5111, Madison, USA) overnight at 37 °C and 1000 rpm and was stopped by adding 18 μl of 2% TFA. Peptide mixtures were desalted using Stop and Go Extraction tips (Rappsilber et al, 2007) self-packed with three layers of $1.0 \times 1.0$ mm AttractSPE® Disks Bio RPS (AFFINISEP, France). Peptides were reconstituted in 0.1% TFA and directly analyzed by LC-MS/MS.

## Mass spectrometry analysis

For LC-MS analysis, an UltiMate™ 3000 RSLCnano system was online coupled to a Q Exactive mass spectrometer (both Thermo Fisher Scientific, Dreieich, Germany). The peptide mixture was washed and preconcentrated on a PepMap C18 trapping column (Thermo Fisher Scientific, Dreieich, Germany) with a flow rate of 30 μl/min and peptides were separated on a μPAC™ C18 pillar array column (200 cm bed length, Thermo Fisher Scientific, Dreieich, Germany) using a binary buffer system consisting of A (0.1% formic acid) and B (86% acetonitrile and 0.1% formic acid). Peptides were eluted applying a linear gradient from 6 to 10% B in 5 min at a flow rate of 0.75 μl/min, 10 to 45% B in 100 min, and 45 to 99% in 8 min at a flow rate of 0.5 μl/min. For electrospray ionization of peptides, a Nanospray Flex ion source (Thermo Fisher Scientific, Dreieich, Germany) with a liquid Junction™ PST-HV-NFU (MS Wil, The Netherlands) and a fused silica emitter (20 μm inner diameter, 360 μm outer diameter, MicroOmics Technologies LLC) applying a source voltage of +1.800 V and an ion transfer tube temperature of 250 °C was used. Cycles of data-dependent acquisition (DDA) consisted of: one MS1 survey scan (RF lens of 50%, normalized AGC target of 3e6, maximum injection time of 60 ms, *m/z* range of 375 to 1.700, resolution of 70.000 at 200 *m/z*,

profile mode) followed by MS2 scans (RF lens of 50%, normalized AGC target of 1e5, maximum injection time of 120 ms, resolution of 35.000, isolation window 3 m/z, exclusion time 45 s, centroid mode) generated for the 12 most abundant peptide signals of the survey scan by higher-energy collision-induced dissociation (HCD) at a normalized energy of 28%.

## MS data analysis

MS raw data were searched using MaxQuant (version 2.0.2.0; (Tyanova et al, 2016)) against the human reference proteome protein sequences from UniProt (release 2023_05) and the sequence of the mouse NLS-Diaph3-wt. Carbamidomethylation of cysteine residues was considered as a fixed modification, oxidation of methionine, acetylation of protein N-termini, as well as serine and threonine phosphorylation were considered as variable modifications. Proteins were identified with at least one unique peptide and a false discovery rate of 0.01 on both the peptide and protein level. The raw data were additionally searched against the same sequences using Mascot (version 2.7.0, www.matrixscience.com) to derive site localization confidences (Savitski et al, 2011).

## Atomic force microscopy (AFM)

All AFM experiments were performed at 37 °C in plastic 35 mm cell culture dishes (TPP) and DMEM medium, buffered with 15 mM HEPES (pH 7.6). Nuclear stiffness was measured with a NanoWizard 4 XP AFM system (JPK) mounted on a Zeiss Axiovert 200 M fluorescence microscope equipped with a Zeiss Colibri 5. Triangular non-conductive Silicon Nitride cantilevers (MLCT-E) with a nominal spring constant $k = 0.1$ N/m were used for the indentation experiments and acquisition of $32 \times 32$ grid force-maps and calibrated using the contact-based method. Nuclei were probed with a force of 5 nN and tip velocity of 10 μm/s at randomly selected positions. The Hertz–Sneddon model was used to fit the force-distance curves in JPK software (Version 8.0.144, JPK Instruments). To estimate the Young's modulus of the nucleus, the average Young's modulus of the four acquired force-distance curves was used. To exclude the possibility of probing the cortex, force-distance curves with indentation depths lower than 1 μm were discarded, and cells expressing NLS-mScarlet-tagged proteins were monitored in the fluorescent channel during the experiments. Expansion of the nucleus during the measurements indicated successful probing of the nucleus.

## Statistical analysis

Statistical analyses were performed using GraphPad 10.1.0 software. Data were presented either as column bar graph ± sd or violin plots with median and quartiles. For comparison of normally distributed data from two groups, a two-tailed parametric $t$-test was performed. For the statistical analysis of more than two groups, one-way analysis of variance (ANOVA) with the corresponding multiple comparison test was applied, as indicated in the figure legends, if the values were normally distributed. Exact numeric $p$ values are indicated in the figures.

## Graphics

BioRender was used to prepare the cartoons of Figs. 1A, 2A, and 3A and a graphical synopsis of this study.

## Data availability

All data were available in the main text or the supplementary materials.

The source data of this paper are collected in the following database record: biostudies:S-SCDT-10_1038-S44318-025-00566-2.

## Peer review information

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

## Acknowledgements

We thank our colleagues for their helpful discussion and comments. This study was supported by Germany's Excellence Strategy (EXC-2189) and the German Research Council (Deutsche Forschungsgemeinschaft) (no. GR2111/15-1 and no. GR2111/13-1). The ELYRA 7 system was funded by a major research instrumentation grant (INST 39/1170-1 FUGG) to RG.

## Author contributions

**Christos Kamaras**: Formal analysis; Validation; Visualization; Methodology; Writing—original draft; Writing—review and editing. **Dennis Frank**: Formal analysis; Investigation; Visualization; Methodology; Writing—review and editing. **Hong Wang**: Investigation; Methodology. **Friedel Drepper**: Investigation; Methodology. **Pitter, F Huesgen**: Investigation; Methodology. **Robert Grosse**: Conceptualization; Supervision; Funding acquisition; Writing—original draft; Project administration; Writing—review and editing.

Source data underlying figure panels in this paper may have individual authorship assigned. Where available, figure panel/source data authorship is listed in the following database record: biostudies:S-SCDT-10_1038-S44318-025-00566-2.

## Funding

## Disclosure and competing interests statement

The authors declare no competing interests.

# Expanded View Figures

**Figure EV1.  Nuclear actin polymerization during confined cell migration is independent of the LINC complex or genetically encoded actin probes.**

(**A**) Representative image sequences of representative HT1080-nAC-GFP cells expressing either mScarlet or the dominant-negative transmembrane domain of nesprin (DN.KASH-mScarlet), migrating through 3 μm microchannels. Actin is visualized via nAC-GFP. Scale bar, 5 μm. (**B**) Percentage of HT1080-nAC-GFP cells displaying nuclear F-actin formation following migration through 3 μm microchannels after co-expression of mScarlet or DN.KASH-mScarlet. Data shown as mean ± s.d. Three independent experiments per condition with $n \geq 6$ per experiment. Statistical analysis was performed with an unpaired parametric *t*-test. ns not significant. (**C**) Image sequences of an HT1080 cell stained with the live-cell actin dye FastActX-SPY650 and expressing NLS-BFP as a NE rupture marker migrating through a microchannel. Scale bar, 5 μm. Before NE rupture; corresponds to the time before a NE rupture event had occurred. NE rupture; corresponds to a frame in which a NE rupture event is detected via leakage of NLS-BFP. (**D**) Image sequences of an HT1080-nAC-GFP cell stained with the live-cell actin dye FastActX-SPY650 migrating through a microchannel. The white arrowhead indicates a nuclear actin filament co-labeled by nAC-GFP and FastActX. Before NE rupture; corresponds to the time before a NE rupture event had occurred. NE rupture; corresponds to a frame in which a NE rupture event is detected via leakage of nAC-GFP. Scale bar, 5 μm. Source data are available online for this figure.

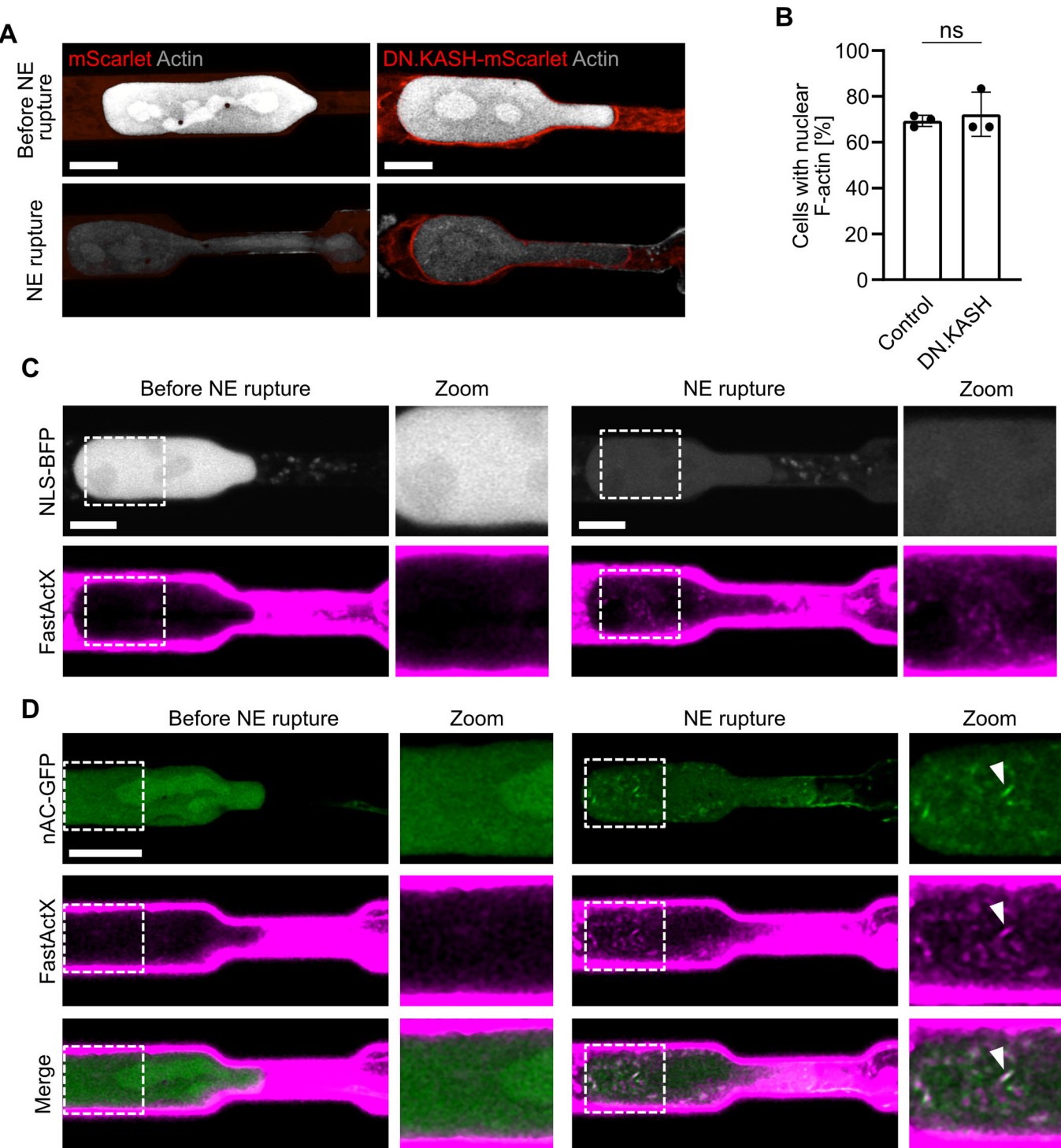

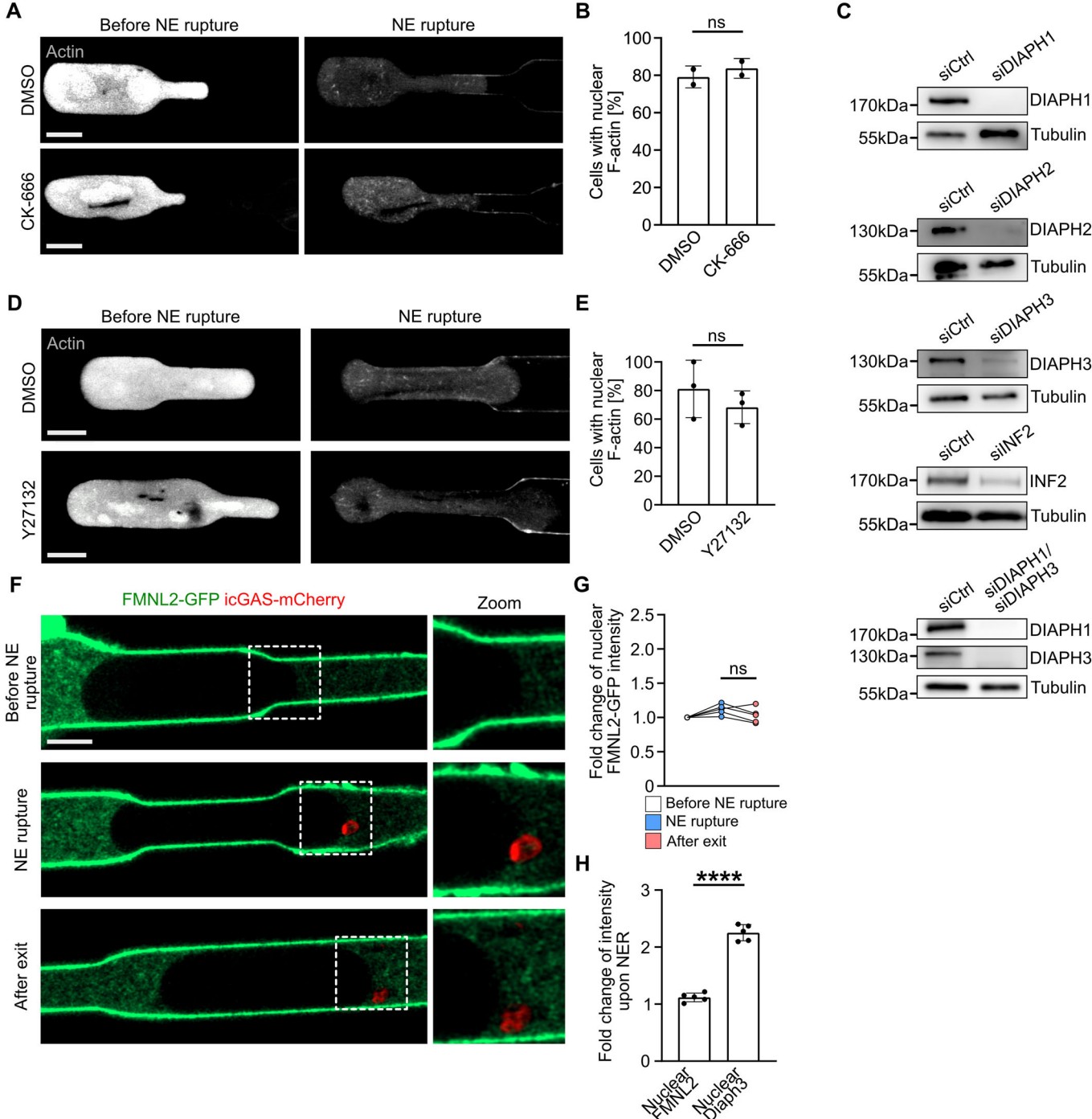

◄ **Figure EV2. Nuclear actin polymerization upon NE rupture is mediated by DIAPH1 and DIAPH3.**

(A) Image sequences of representative HT1080-nAC-GFP cells migrating through 3 μm microchannels, after being treated with either 0.01% DMSO or 100 μM of the Arp-2/3 inhibitor CK-666. Actin is visualized via nAC-GFP. Scale bar, 5 μm. (B) Percentage of HT1080-nAC-GFP displaying nuclear F-actin structures after treatment with either 0.01% DMSO or 100 μM CK-666. Data shown as mean ± s.d. Two independent experiments per condition. $n \geq 8$ cells per independent experiment. (C) Representative immunoblots for the silencing efficiency of DIAPH1 (siDIAPH1), DIAPH2 (siDIAPH2), DIAPH3 (siDIAPH3), DIAPH1 and DIAPH3 (siDIAPH1/siDIAPH3), and INF2 (siINF2). Tubulin serves as a loading control. (D) Image sequences of HT1080-nAC-GFP cells that were treated with 0.01% DMSO or ROCK inhibitor 10 μM Y27132 before and during NE rupture. Actin is visualized via nAC-GFP. Scale bar, 5 μm. (E) Percentage of cells displaying nuclear F-actin upon NE rupture after being treated with either DMSO or Y27132. Data shown as mean ± s.d. Three independent experiments per condition with $n \geq 6$ cells per experiment. Statistical analysis was performed with an unpaired parametric $t$-test. (F) Image sequences of a representative HT1080-icGAS-mCherry cell that expresses FMNL2-GFP migrating through a microchannel. Before NE rupture; corresponds to the time before a NE rupture event had occurred. NE rupture; corresponds to the first frame in which a NE rupture event is detected via accumulation of icGAS-mCherry. After exit; corresponds to the time point at which the nucleus has exited from the narrow part of the microchannel. Scale bar, 5 μm. (G) Quantification of nuclear FMNL2-GFP intensity before NE rupture, during NE rupture and after exit. Data shown as individual values from $n = 5$ cells. Statistical analysis was performed by a paired parametric $t$-test. (H) Fold change of nuclear FMNL2-GFP and nuclear GFP-Diaph3 mean fluorescence intensities upon NE rupture. Data shown as individual values from $n = 5$ cells. Statistical analysis was performed by an unpaired parametric $t$-test. Exact $P$ value: $3.0 \times 10^{-7}$. ****$P < 0.0001$ and ns not significant. Source data are available online for this figure.

**A**

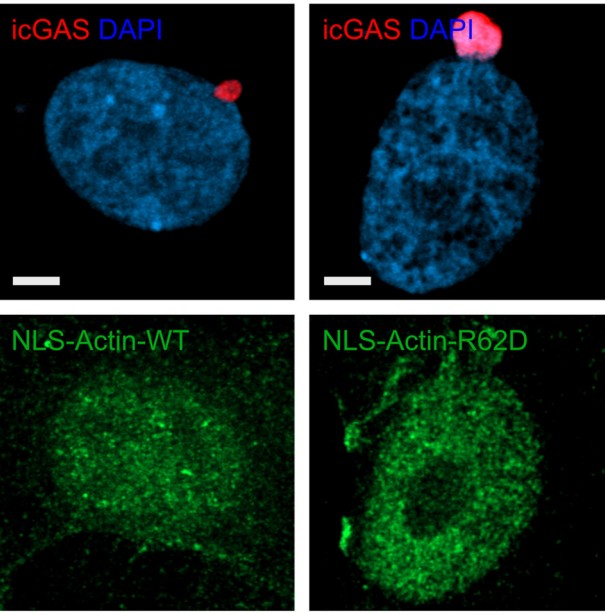

Figure EV3.  **Nuclear actin polymerization maintains nuclear integrity during confined cell migration.**

(**A**) Representative immunofluorescence images of HT1080 cells stably expressing icGAS-mCherry (red) that transiently express either NLS-Actin-WT or NLS-Actin-R62D (green). These images were reused from Fig. 4D to show the expression of NLS-actin-WT or NLS-actin-R62D in these examples. Scale bar, 5 μm. Source data are available online for this figure.

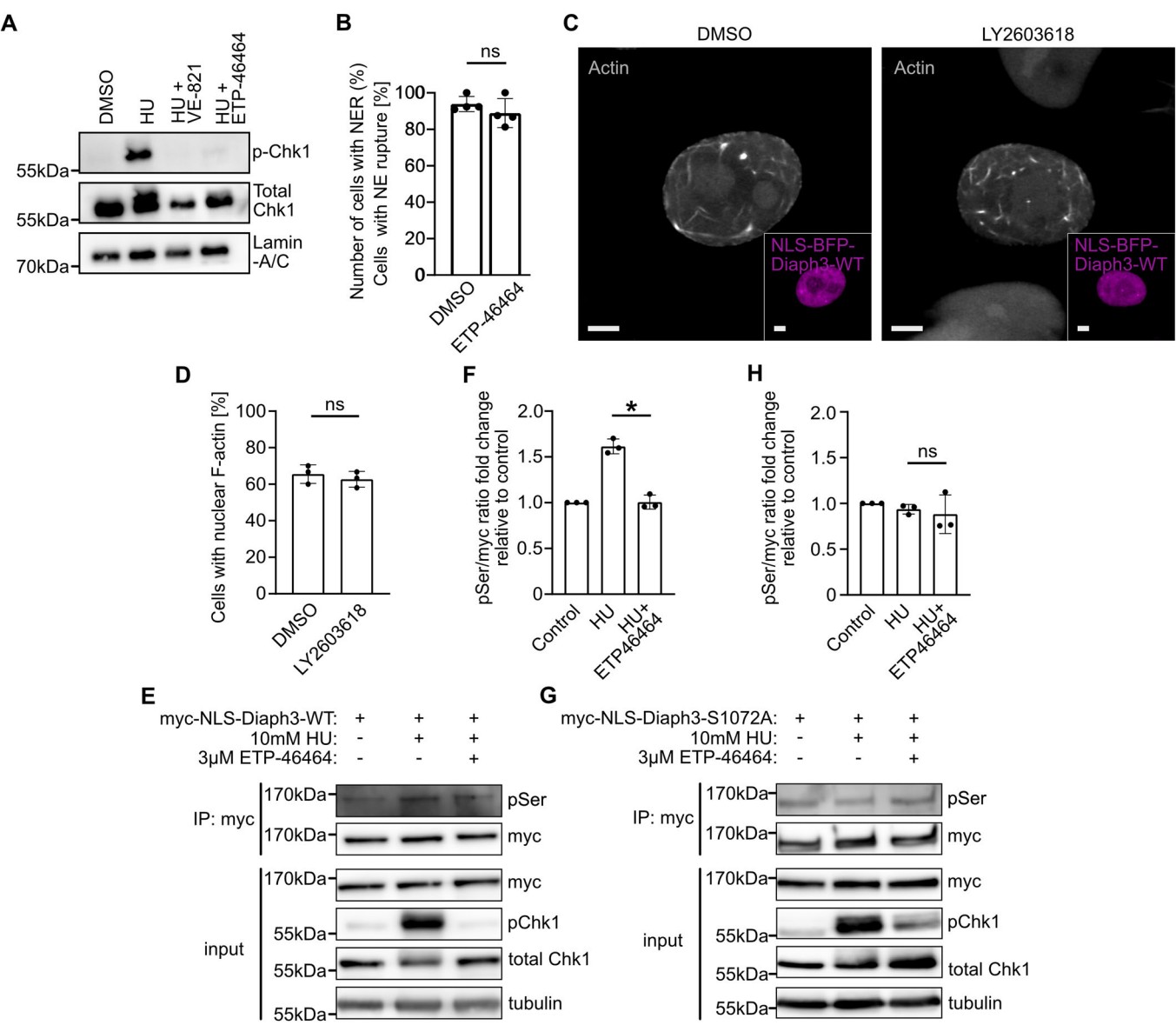

**Figure EV4. ATR is necessary for nuclear actin polymerization upon NE rupture.**

(A) Representative immunoblot of phosphorylated Chk1 (pChk1) (stripped and reprobed for total-Chk1) for the efficiency of ATR inhibitors ETP-46464 and VE-821. Loading control Lamin-A/C. (B) Percentage of HT1080 cells undergoing NE rupture after being treated with siCtrl or siATR. Data shown as mean ± s.d. Four independent experiments per condition with $n \geq 8$ cells per experiment. Statistical analysis was performed with the Mann–Whitney test. ns not significant. (C) Representative images of HT1080-nAC-GFP (white) that transiently express NLS-BFP-Diaph3-wt in the presence or absence of LY2603618, with bottom right images showing nuclear expression of this construct (magenta); Scale bar, 4 µm. (D) Percentage of HT1080-nAC-GFP cells transiently expressing NLS-Diaph3-wt that display nuclear F-actin formation after being treated with either 0.01% DMSO or 1 µM LY2603618 for 16 h. Data shown as mean ± s.d. Three independent experiments per condition with $n = 30$ cells per experiment. Statistical analysis between DMSO and LY2603618 treatment after expression of NLS-BFP-Diaph3-WT was performed by an unpaired parametric $t$-test. (E) HT1080-WT cells were transfected with myc-NLS-Diaph-WT, and cells were treated with Hydroxyurea (HU) in the presence or absence of ETP-46464. Immunoprecipitation was carried out by myc-agarose beads. The amount of phosphorylated serine residues was determined via western blot and detected with a general anti-phosphoserine antibody (pSer) (stripped and reprobed for myc). pChk1 (stripped and reprobed for total-Chk1 and Tubulin) was used as a marker for ATR activation and to monitor the activity of ETP-46464. (F) Quantification of the phosphorylation of serine residues of myc-NLS-BFP-Diaph3-WT upon HU treatment in the presence or absence of ETP-46464. Fold change of the pSer/myc ratio was calculated. Data shown as mean ± s.d. Three independent experiments per condition. Statistical analysis was performed with a paired parametric $t$-test. Exact $P$ value: 0.01948. (G) HT1080-WT cells were transfected with myc-NLS-Diaph-S1072A, and cells were treated with HU in the presence or absence of ETP-46464. Immunoprecipitation was carried out by myc-agarose beads. The amount of phosphorylated serine residues was determined via western blot and detected with a general anti-phosphoserine antibody (pSer) (stripped and reprobed for myc). pChk1 (stripped and reprobed for total-Chk1 and Tubulin) was used as a marker for ATR activation and to monitor the activity of ETP-46464. (H) Quantification of the phosphorylation of serine residues of myc-NLS-BFP-Diaph3-S0172A upon HU treatment in the presence or absence of ETP-46464. Fold change of the pSer/myc ratio was calculated. Data shown as mean ± s.d. Three independent experiments per condition. Statistical analysis was performed with a paired parametric $t$-test. *$P < 0.05$ and ns was not significant. Source data are available online for this figure.

