## [Peer Review File · The EMBO Journal]

Nuclear rupture in confined cell migration triggers nuclear actin assembly to limit chromatin leakage

Christos Kamaras, Dennis Frank, Hong Wang, Friedel Drepper, Pitter Huesgen, and Robert Grosse

Corresponding authors: Robert Grosse (robert.grosse@pharmakol.uni-freiburg.de) , Christos Kamaras (christos.kamaras@pharmakol.uni-freiburg.de)

Review Timeline:

Submission Date:	12th Apr 25
Editorial Decision:	15th May 25
Revision Received:	21st Jul 25
Editorial Decision:	19th Aug 25
Revision Received:	27th Aug 25
Accepted:	29th Aug 25

Editor: Ieva Gailite

Transaction Report:

Dear Dr. Grosse,

Thank you for submitting your manuscript for consideration by the EMBO Journal. We have now received comments from a full set of reviewers, which are included below for your information.

As you will see, all reviewers are generally positive in their assessment and appreciate the contribution of the study to the research field. At the same time, they indicate several concerns that would be important to address in the revised study. In particular, reviewer #1 asks to provide further analysis to rule out the canonical function of ATR, while both reviewers #1 and #3 request to provide further substantiation of ATR-dependent phosphorylation of DIAPH3. While the farther-reaching analyses requested by reviewer #3 in the first major point will not be required, clarification of the contributions of other reported pathways of ATR-dependent actin polymerisation as requested by reviewer #2 would substantially contribute to the study. Taken together, I invite you to submit a revised manuscript in which you address the reviewers' concerns along the lines outlined above. I think that it would be useful to discuss the revision in more detail via email or phone/videoconferencing - please let me know which option you prefer.

We generally allow three months as standard revision time, which can be extended to six months in the case of major revisions. Should you foresee a problem in meeting this deadline, please let us know in advance to discuss an extension. As a matter of policy, competing manuscripts published during this period will not negatively impact on our assessment of the conceptual advance presented by your study. However, please contact me as soon as possible upon publication of any related work to discuss the appropriate course of action.

When preparing your letter of response to the referees' comments, please bear in mind that this will form part of the Review Process File and will therefore be available online to the community. For more details on our Transparent Editorial Process, please visit our website: <https://www.embopress.org/page/journal/14602075/authorguide#transparentprocess>. Please also see the attached instructions for further guidelines on preparation of the revised manuscript.

Please feel free to contact me if you have any further questions regarding the revision. Thank you for the opportunity to consider your work for publication. I look forward to discussing your revision.

With best regards,

leva

leva Gailite, PhD
Senior Scientific Editor
The EMBO Journal
Meyerhofstrasse 1
D-69117 Heidelberg
Tel: +4962218891309
i.gailite@embojournal.org

- a point-by-point response to the referees' comments, with a detailed description of the changes made (as a word file).
- a word file of the manuscript text.
- individual production quality figure files (one file per figure)
- a complete author checklist, which you can download from our author guidelines

(<https://www.embopress.org/page/journal/14602075/authorguide>).

- Expanded View files (replacing Supplementary Information)

- a Reagents and Tools Table as part of the Methods section, which can be downloaded from our author guidelines

(<https://www.embopress.org/page/journal/14602075/authorguide#structuredmethods>)

We realize that it is difficult to revise to a specific deadline. In the interest of protecting the conceptual advance provided by the work, we recommend a revision within 3 months (13th Aug 2025). Please discuss the revision progress ahead of this time with the editor if you require more time to complete the revisions.

Referee #1:

The manuscript by Karmas et al. covers the novel finding that metastasising or migrating cells protect their genome from excessive DNA damage and chromosome loss. In response to inevitable nuclear envelope rupture that occurs as a consequence of cells crawling through confined spaces, ATR appears to phosphorylate the forming DIAPH3 which is responsible for generating nuclear actin structures. These Nuclear actin filaments create a stiff nucleus which protects from chromatin leakage. The study is clear well written and the data supports the conclusions. The following comments are suggestions to help clarify the general mechanism and wider importance.

Comments.

1. NER is an existing abbreviation in the DNA damage field (nucleotide excision repair) so it may be wise to alter this to avoid confusion.
2. In Kumar et al. and Kidiyoor et al. the authors demonstrated NE recruitment of ATR in response to mechanical tension - is the same true here? They used HeLa and HCT116, the work here focuses on HT1080 so would be good to bring a holistic view if possible as cells may have different mutations impacting generalisability.
3. Could the authors comment on the peri-nuclear DIAPH3 staining pre-nuclear env breakdown. Does ATR phosphorylation of DIAPH3 also occur under osmotic stress/mechanical stretching or simply stiff ECM?
4. Could the authors possibly include a CHK1 inhibitor to support an ATR-DIAPH3 role and rule out canonical ATR DDR signalling.
5. The statement 'These data demonstrate that ATR kinase controls nuclear rigidity by regulating formin-mediated actin assembly phosphorylation' is a little strong as the study is not definitive. Consider using 'ATR kinase CAN control nuclear'.
6. The statement 'and only very recently was proposed to promote nuclear actin polymerization upon osmotic stress via recruitment of Filamin-A to the NE through binding with RASS1FA' is not correct 'has been shown to promote nuclear actin polymerization upon osmotic and mechanical stress via recruitment of Filamin-A to the NE through binding with ATR-phosphorylated RASSF1A' is more appropriate.

Referee #2:

The manuscript by Kamaras et al describes a novel signaling mechanisms culminating in nuclear actin polymerization to promote chromatin integrity during confined migration.

In tissues, cells encounter significant mechanical challenges during migration through extracellular matrix and the ability to survive constricted migration plays a significant role for example during cancer metastasis. Constricted migration can result in nuclear envelope rupture (NER), compromising the barrier between cytoplasm and nucleus, and making the genome vulnerable to damage. It is thus important to understand the molecular mechanisms that the cell utilizes to overcome these threats to genomic integrity.

In the present manuscript, the authors convincingly demonstrate dynamic assembly of nuclear actin filaments, dependent on DIAPH1/3, during confined migration. NER seems to precede nuclear actin assembly, and is linked to nuclear accumulation of

DIAPH3, a formin protein that can promote actin polymerization. The authors further show that ATR phosphorylates (directly or indirectly) DIAPH3, and that ATR-DIAPH3 dependent nuclear actin polymerization generally affects nuclear stiffness, and is required to prevent chromatin leakage during confined migration. This data thus adds to recent reports linking ATR induced nuclear actin polymerization to mechanical stimulation (Chatzifrangkeskou et al., 2025) and replication stress (Lamm et al., 2020).

Overall, this is an interesting and carefully conducted study that provides a compelling pathway for protecting the genome during cell migration and providing yet another functional role for nuclear actin dynamics. Nevertheless, some aspects should be clarified before publication.

Major points

The role of nuclear envelope rupture (NER) in the process described here remains somewhat unclear. According to the model, it seems that NER would be required for nuclear accumulation of DIAPH3. However, the experiments supporting this part would require further controls to provide specificity for this phenomenon, since NER is expected to generally lead to mixing of cytoplasmic and nuclear contents. How is NER affecting the subcellular distribution of DIAPH2 that does not seem to be required for nuclear actin assembly in this context. Moreover, studies with RASSF1 have suggested that ATR would operate especially on the nuclear envelope upon mechanical stimuli (Chatzifrangkeskou et al., 2025). Here it seems that DIAPH3 is lost from the NE upon NER; is this true? Also, previous studies had suggested that upon replication stress, ATR induces nuclear actin polymerization via Arp2/3. Some discussion on how and why the same upstream regulator (ATR) operates via two distinct mechanisms (Arp2/3 vs. formins) to achieve the same end result (nuclear actin polymerization) for overall the same purpose (protecting the genome) would be very interesting.

In figure 3, it is unclear how the ability to migrate in general is related to the nuclear fragmentation or vice versa. It seems that in some of the videos with perturbed nuclear actin polymerization the cell is stuck, and then the nucleus is fragmented, whereas the schematic in fig 3 implies the other way around.

Minor points

It would be good to use the same nomenclature in both graphical synopsis and in rest of the manuscript (DIAPH3 vs mDia2). Since ATR has been linked to nuclear actin polymerization already before, these papers would be appropriate to mention already in the introduction to provide proper context.

It is an overstatement to conclude on page 4, lines 139-140 that NER triggers nuclear actin polymerization. The experiment simply investigates their temporal, not functional relationship.

Chatzifrangkeskou, M., T. Stanly, D. Koennig, L. Campos-Soares, M. Eyres, A. Hasson, A. Perdiou, I. Vendrell, R. Fischer, S. Das, S. Gardner, S. Go, B. Fletcher, A. Newton, P. Skourides, F. Szele, and E. O'Neill. 2025. ATR-hippo drives force signaling to nuclear F-actin and links mechanotransduction to neurological disorders. *Sci Adv.* 11:eadr5683.

Lamm, N., M.N. Read, M. Nobis, D. Van Ly, S.G. Page, V.P. Masamsetti, P. Timpson, M. Biro, and A.J. Cesare. 2020. Nuclear F-actin counteracts nuclear deformation and promotes fork repair during replication stress. *Nat Cell Biol.* 22:1460-1470.

Referee #3:

The manuscript by Kamaras and co-workers identifies a role for nuclear actin polymerization in preventing excessive nuclear rupture and chromatin herniation upon HT-1080 migration through narrow pores. They demonstrate that nuclear actin forms at the rear end of nuclei through activity of DIAPH3 that enters the nucleus during the deformation of the nucleus within a narrow pore. They further demonstrate that this effects involves ATR activity and phosphorylation of DIAPH3, that relate with nuclear stiffness, that through unknown mechanisms prevents excessive nuclear rupture.

All in all this is a very interesting manuscript and the data is mostly convincing. The process that the authors describe is new and will be of interest for the cell biology community. While the mechanism of nuclear actin polymerization is quite well described, the mechanism by which it regulates nuclear stiffness and nuclear rupture remains unclear, especially given that the polymerization does not occur at the site of the rupture. Elucidating this mechanism is likely to be a substantial effort but at least addressing some of the previously published mechanisms leading to nuclear rupture, most importantly recruitment of the ESCRT complex, levels of laminA and laminB receptor, could be analyzed here.

In addition to this major point, there are two additional points that should be addressed prior to publication

1. The nuclear actin data is based entirely on cells expressing nuclear actin chromobody. While this data is clear, it is well known that expression actin reporters can change the kinetics/stoichiometry of actin polymerization and trigger actin polymerization. The authors should show actin filament formation in cells that do not express the chromobody.

2. The authors claim that DIAPH3 is phosphorylated by ATR. While ATR inhibition blocks nuclear actin polymerization, there is no evidence that DIAPH3 is an ATR substrate or that ATR is activated by nuclear deformation. The authors should demonstrate ATR activation as well as show, preferentially using *In vitro* kinase assays that DIAPH3 is indeed a substrate of ATR. At minimum, they should show that DIAPH3 phosphorylation is decreased upon ATR inhibition, which would not be sufficient to

claim that it is an ATR substrate but at least demonstrate that ATR activity is impacting (directly or indirectly) DIAPH3 phosphorylation.

point-by-point response

We thank the referees and editors for their time and effort to assess our work and in particular for their helpful and constructive criticism. We were happy about the overall positive feedback and that our study is "clear", "novel" and "convincing", "interestingly and carefully conducted", and "very interesting and mostly convincing". We have made an effort to address all remaining points within the provided revision deadline and hope that we were able to answer the referees' comments and suggestions to their satisfaction.

Reviewer #1

The manuscript by Karmas et al. covers the novel finding that metastasising or migrating cells protect their genome from excessive DNA damage and chromosome loss. In response to inevitable nuclear envelope rupture that occurs as a consequence of cells crawling through confined spaces, ATR appears to phosphorylate the forming DIAPH3 which is responsible for generating nuclear actin structures. These Nuclear actin filaments create a stiff nucleus which protects from chromatin leakage. The study is clear and well written and the data supports the conclusions. The following comments are suggestions to help clarify the general mechanism and wider importance.

1. NER is an existing abbreviation in the DNA damage field (nucleotide excision repair) so it may be wise to alter this to avoid confusion.

We agree with Reviewer #1 and have changed all instances of "NER (nuclear envelope rupture)" to "nuclear envelope rupture (NE rupture)" throughout the manuscript to avoid confusion with nucleotide excision repair

2. In Kumar et al. and Kidiyoor et al. the authors demonstrated NE recruitment of ATR in response to mechanical tension - is the same true here? They used HeLa and HCT116, the work here focuses on HT1080 so would be good to bring a holistic view if possible as cells may have different mutations impacting generalisability.

We thank the reviewer for pointing this out. To our knowledge, there is no reported mutation of ATR in HT1080 cells. Further to that, in the paper by Kumar et al. they used immunostaining for endogenous ATR (Kumar et al., 2014). Unfortunately, immunostaining of cells inside the microchannels is not possible. Therefore, we cannot repeat the same experiment in our specific setup. We apologize for this. Nevertheless, we addressed this point experimentally by performing immunostainings of HT1080 for endogenous ATR upon osmotic stress as previously described (Chatzifrangkeskou et al., 2025; Kumar et al.,

2014). We were able to show that in our cell line, ATR is able to translocate to the NE in response to osmotic stress (see reviewer figure 1).

Reviewer figure 1. ATR localizes to the nuclear envelope upon osmotic stress. Endogenous immunostaining of ATR and Lamin A/C in HT1080 cells. (A) Control cells display homogeneous distribution of ATR in the nucleoplasm. Lower panel shows line-scan at indicated plane (dashed line in zoom). Scale bar, 10 μm . Scale bar for Zoom, 4 μm . (B) ATR accumulates at the nuclear envelope. Lower panel shows line-scan at indicated plane (dashed line in zoom). Black arrows in line-scan indicate localization of ATR at the nuclear envelope as indicated by the staining of Lamin A/C. Scale bar, 10 μm . Scale bar for Zoom, 4 μm .

3. Could the authors comment on the peri-nuclear DIAPH3 staining pre-nuclear envelope breakdown. Does ATR phosphorylation of DIAPH3 also occur under osmotic stress/mechanical stretching or simply stiff ECM?

According to a paper from the lab of Alexander Bershadsky (Shao et al., 2015) both endogenous and overexpressed Diaph3 localizes to the perinuclear, cytoplasmic side of the NE, which is consistent with our data prior to NE rupture during confined migration.

Regarding the comment on whether ATR-dependent phosphorylation of DIAPH3 also occurs upon osmotic stress, we had indeed included sorbitol treated cells in the presence and absence of the ATR inhibitor ETP-46464 in our

phosphoproteomic analysis back then and in fact we did observe an increase of phosphorylation of DIAPH3 in the (QSLS*PMSQRPVLK) peptide following sorbitol treatment, which was diminished upon pharmacological inhibition of ATR (see reviewer figure 2).

Reviewer figure 2. ATR regulates the phosphorylation of Diaph3 at S1072 upon osmotic stress.

Fold change of detected phosphorylation of nuclear Diaph3 in the (QSLS*PMSQRPVLK) peptide normalized to control upon treatment with 0.5M Sorbitol in the presence or absence of ATR inhibitor ETP-46464.

4. *Could the authors possibly include a CHK1 inhibitor to support an ATR-DIAPH3 role and rule out canonical ATR DDR signalling.*

This is an important point raised by reviewer #1. We addressed this point experimentally using the specific Chk1 inhibitor LY2603618. We found that inhibition of Chk1 kinase activity had no effect on nuclear F-actin formation now shown in Figs. EV4C and D. These results indicate that ATR-mediated regulation of Diaph3 is independent of the canonical ATR-DNA damage response pathway. We have incorporated these new data and updated the discussion in the revised manuscript.

We thank the reviewer for pointing this out.

5. *The statement 'These data demonstrate that ATR kinase controls nuclear rigidity by regulating formin-mediated actin assembly phosphorylation' is a little strong as the study is not definitive. Consider using 'ATR kinase CAN control nuclear'.*

We agree. We have now revised this to: "These data suggest that ATR kinase can influence nuclear rigidity by regulating formin-mediated actin assembly." to reflect our findings more accurately.

6. The statement 'and only very recently was proposed to promote nuclear actin polymerization upon osmotic stress via recruitment of Filamin-A to the NE through binding with RASSIFA' is not correct 'has been shown to promote nuclear actin polymerization upon osmotic and mechanical stress via recruitment of Filamin-A to the NE through binding with ATR-phosphorylated RASSF1A' is more appropriate.

We thank Reviewer #1 for the clarification. We have corrected the sentence accordingly on page 7 lines 274-277.

Reviewer #2:

The manuscript by Kamaras et al describes a novel signaling mechanisms culminating in nuclear actin polymerization to promote chromatin integrity during confined migration.

In tissues, cells encounter significant mechanical challenges during migration through extracellular matrix and the ability to survive constricted migration plays a significant role for example during cancer metastasis. Constricted migration can result in nuclear envelope rupture (NER), compromising the barrier between cytoplasm and nucleus, and making the genome vulnerable to damage. It is thus important to understand the molecular mechanisms that the cell utilizes to overcome these threats to genomic integrity.

In the present manuscript, the authors convincingly demonstrate dynamic assembly of nuclear actin filaments, dependent on DIAPH1/3, during confined migration. NER seems to precede nuclear actin assembly, and is linked to nuclear accumulation of DIAPH3, a formin protein that can promote actin polymerization. The authors further show that ATR phosphorylates (directly or indirectly) DIAPH3, and that ATR-DIAPH3 dependent nuclear actin polymerization generally affects nuclear stiffness, and is required to prevent chromatin leakage during confined migration. This data thus adds to recent reports linking ATR induced nuclear actin polymerization to mechanical stimulation (Chatzifrangkeskou et al., 2025) and replication stress (Lamm et al., 2020).

Overall, this is an interesting and carefully conducted study that provides a compelling pathway for protecting the genome during cell migration and providing yet another functional role for nuclear actin dynamics. Nevertheless, some aspects should be clarified before publication.

The role of nuclear envelope rupture (NER) in the process described here remains somewhat unclear. According to the model, it seems that NER would be required for nuclear accumulation of DIAPH3. However, the experiments supporting this part would require further controls to provide specificity for this phenomenon, since NER is expected to generally lead to mixing of cytoplasmic and nuclear contents. How is NER affecting the subcellular distribution of DIAPH2 that does not seem to be required for nuclear actin assembly in this context.

We thank the reviewer for this important critique. Since DIAPH2 plasmids were unavailable to us we addressed this by examining the localization dynamics of the closely related Diaphanous-related formin FMNL2 (Wang et al., 2015). Notably, unlike Diaph3, FMNL2-GFP did not accumulate in the nucleus upon NE rupture and remained cytoplasmic (Fig EV2 F-H, Movie EV8). We have now added these new control experiments in the revised manuscript in Figure Figs. EV2F-H showing specificity for nuclear accumulation of Diaph3.

Moreover, studies with RASSF1 have suggested that ATR would operate especially on the nuclear envelope upon mechanical stimuli (Chatzifrangkeskou et al., 2025). Here it seems that DIAPH3 is lost from the NE upon NE rupture; is this true?

According to a study from the lab of Alexander Bershadsky (Shao et al., 2015), DIAPH3 is located at the outer nuclear membrane, which appears to be consistent with our data (Fig. 2C). In our experiments, we observed that upon NE rupture, Diaph3 rapidly accumulates within the nuclear interior while perinuclear localization becomes diminished during this process but is eventually reversed (Fig. 2C). The precise molecular mechanism for rapid nuclear accumulation of DIAPH3 upon NE rupture remains to be examined in more detail in the future. Nevertheless, the perinuclear localization of DIAPH3 may predestine it for rapid nuclear translocation.

Also, previous studies had suggested that upon replication stress, ATR induces nuclear actin polymerization via Arp2/3. Some discussion on how and why the same upstream regulator (ATR) operates via two distinct mechanisms (Arp2/3 vs. formins) to achieve the same end result (nuclear actin polymerization) for overall the same purpose (protecting the genome) would be very interesting.

While in both cases nuclear actin appears to be ATR dependent, it has been shown that ATR can be activated by both replication stress (canonical DDR pathway) and mechanical tension. Although the end goal in both situations is genome protection, the trigger as well as actin filament architecture are markedly different. During replication stress, Lamm et al. reported that nuclear F-actin persists in some form of actin cables for multiple hours to facilitate the repair of stalled replication forks and to allow for exit from the S-phase and cell cycle progression (Lamm et al., 2020). It should be noted that in these studies on replication stress-induced actin, no evidence regarding the actual presence of Arp2/3 in the nucleus or at replication stress sites could be provided, while conclusions mostly rely on the global inhibition of Arp2/3 using CK666. Here, using live cell imaging we unambiguously demonstrate that Diaph3 accumulates rapidly in the nucleus to facilitate nuclear actin polymerization for mechanical support of the nuclear organelle.

In our study, the formin-mediated filaments appear to be very distinct, short, thin and very dynamic and transient, while the replication stress induced

filaments in Lamm et al. 2020 appear to be more stable and thick persisting for hours on end.

These findings highlight that, although in both of these scenarios ATR converges on nuclear actin polymerization for genome protection, the upstream trigger and downstream nuclear actin architectures are very different.

As suggested, we now added a paragraph on page 7-8, lines 296-304 in the Discussion to address this.

In figure 3, it is unclear how the ability to migrate in general is related to the nuclear fragmentation or vice versa. It seems that in some of the videos with perturbed nuclear actin polymerization the cell is stuck, and then the nucleus is fragmented, whereas the schematic in fig 3 implies the other way around.

We thank the reviewer for raising this point. Our labeling “halted migration” in the cartoon in figure 3A was perhaps misleading and incorrect and is now changed into “fragmented nucleus”. Also, although occasionally cells pause migrating through the constrictions, this rarely causes cell death as the majority of the cells eventually continue to migrate. As an example, we kindly refer to the time-lapse figure provided below. (Reviewer figure 3)

Reviewer figure 3. Cells that are stuck retain their migratory capacity.

Image sequence of an HT1080 cell expressing H2B-mCherry (red), icGAS-GFP (green) and NLS-BFP-Actin-R62D (blue) migrating through 3µm microchannels experiencing NE rupture, getting stuck in the narrow part of the microchannel while retaining its ability to further migrate (white box). White dashed lines indicate the microchannel walls. Scale bar, 10µm.

Minor:

It would be good to use the same nomenclature in both graphical synopsis and in rest of the manuscript (DIAPH3 vs mDia2).

To avoid any confusion to the readers we have changed all instances of mDia2 to Diaph3 on the graphical synopsis.

Since ATR has been linked to nuclear actin polymerization already before, these papers would be appropriate to mention already in the introduction to provide proper context.

We now modified the Introduction section accordingly on page 3 line 101-103.

It is an overstatement to conclude on page 4, lines 139-140 that NER triggers nuclear actin polymerization. The experiment simply investigates their temporal, not functional relationship.

We rephrased this line accordingly stating that NE rupture precedes nuclear actin polymerization.

Chatzifrangkeskou, M., T. Stanly, D. Koennig, L. Campos-Soares, M. Eyres, A. Hasson, A. Perdiou, I. Vendrell, R. Fischer, S. Das, S. Gardner, S. Go, B. Futcher, A. Newton, P. Skourides, F. Szele, and E. O'Neill. 2025. ATR-hippo drives force signaling to nuclear F-actin and links mechanotransduction to neurological disorders. *Sci Adv.* 11:eadr5683.

Lamm, N., M.N. Read, M. Nobis, D. Van Ly, S.G. Page, V.P. Masamsetti, P. Timpson, M. Biro, and A.J. Cesare. 2020. Nuclear F-actin counteracts nuclear deformation and promotes fork repair during replication stress. *Nat Cell Biol.* 22:1460-1470.

Reviewer #3:

The manuscript by Kamaras and co-workers identifies a role for nuclear actin polymerization in preventing excessive nuclear rupture and chromatin herniation upon HT-1080 migration through narrow pores. They demonstrate that nuclear actin forms at the rear end of nuclei through activity of DIAPH3 that enters the nucleus during the deformation of the nucleus within a narrow pore. They further demonstrate that this effects involves ATR activity and phosphorylation of DIAPH3, that relate with nuclear stiffness, that through unknown mechanisms prevents excessive nuclear rupture.

All in all this is a very interesting manuscript and the data is mostly convincing. The process that the authors describe is new and will be of interest for the cell biology community.

While the mechanism of nuclear actin polymerization is quite well described, the mechanism by which it regulates nuclear stiffness and nuclear rupture remains unclear, especially given that the polymerization does not occur at the site of the rupture. Elucidating this mechanism is likely to be a substantial effort but at least addressing some of the previously published mechanisms leading to nuclear rupture, most importantly recruitment of the ESCRT complex, levels of laminA and laminB receptor, could be analyzed here.

We thank reviewer #3 for the recommendations. While we agree that investigating whether impaired nuclear actin polymerization also affects NE repair and the composition of NE would provide important mechanistic insights, we think that such comprehensive analysis may be perhaps beyond the scope of this study. Nevertheless, we did try some initial experiments by expressing CHMP4B-mCherry (Addgene #116923) on cells undergoing confined migration treated with either siCtrl or siDIAPH1/3. Unfortunately, no transfected cells were not able to initiate migration into the microchannels, suggesting that expression of this construct was potentially harmful to the cells, at least in our experimental setup. We will try to address this further in future studies.

1. In addition to this major point, there are two additional points that should be addressed prior to publication. The nuclear actin data is based entirely on cells expressing nuclear actin chromobody. While this data is clear, it is well known that expression actin reporters can change the kinetics/stoichiometry of actin polymerization and trigger actin polymerization. The authors should show actin filament formation in cells that do not express the chromobody.

This is an important point. While endogenous nuclear F-actin is commonly visualized by phalloidin staining, analysis of fixed cells inside the microchannels is currently not feasible. Nevertheless, we addressed this using the novel live-cell imaging actin dye FastActX-SPY650 and observed the formation of short and dynamic nuclear F-actin structures, following NE rupture (by leakage of NLS-BFP to the cytoplasm). We now include these important controls as new data in (Movie EV3 and 4) and the corresponding images (Fig EV1C and D) to the revised manuscript.

2. The authors claim that DIAPH3 is phosphorylated by ATR. While ATR inhibition blocks nuclear actin polymerization, there is no evidence that DIAPH3 is an ATR substrate or that ATR is activated by nuclear deformation. The authors should demonstrate ATR activation as well as show, preferentially using In vitro kinase assays that DIAPH3 is indeed a substrate of ATR. At minimum, they should show that DIAPH3 phosphorylation is decreased upon ATR inhibition, which would not be sufficient to claim that it is an ATR substrate but at least demonstrate that ATR activity is impacting (directly or indirectly) DIAPH3 phosphorylation.

This is a valid and important point of criticism raised by reviewer #3. To address this, we purified GST-Diaph3-CT (amino-acids 1036-1171) subjected it to a commercially available kinase assays (Cat. #14-953) containing a

“recombinant ATR/ATRIP complex”. Unfortunately, our attempts using the only commercially available ATR *in-vitro* kinase assay during the revision period were unsuccessful. This could be due to various and several reasons, which would require extensive trouble shooting that is further complicated by the fact that there is very little literature reporting on non-radioactive ATR kinase assays, in fact there is only one paper (Achuthankutty et al., 2019). Hence, to establish, validate and optimize this assay to yield reproducible results would take an unforeseeable amount of time.

To better address the issue of ATR involvement as suggested by the reviewer, we performed immunoprecipitation experiments on myc-NLS-BFP-Diaph3-wt and myc-NLS-BFP-Diaph3-S1072A after HU treatment in the presence or absence of a pharmacological ATR inhibitor. Using an anti-phosphoserine antibody, we found that NLS-Diaph3-wt displayed an increase in phosphorylated serine upon HU treatment, which was reduced upon ATR inhibition. In contrast, NLS-Diaph3-S1072A displayed no detectable changes. These new data now shown as in Figs. EV4E-H. further support the notion that ATR impacts Diaph3 phosphorylation. Since this does not fully evidence direct phosphorylation by ATR, we toned down our wording accordingly in the manuscript to accurately reflect our findings throughout the manuscript.

- Achuthankutty, D., Thakur, R. S., Haahr, P., Hoffmann, S., Drainas, A. P., Bizard, A. H., Weischenfeldt, J., Hickson, I. D., & Mailand, N. (2019). Regulation of ETAA1-mediated ATR activation couples DNA replication fidelity and genome stability. *Journal of Cell Biology*, *218*(12), 3943–3953. <https://doi.org/10.1083/JCB.201905064>
- Chatzifrangkeskou, M., Stanly, T., Koennig, D., Campos-Soares, L., Eyres, M., Hasson, A., Perdiou, A., Vendrell, I., Fischer, R., Das, S., Gardner, S., Go, S., Futcher, B., Newton, A., Skourides, P., Szele, F., & O’neill, E. (2025). ATR-hippo drives force signaling to nuclear F-actin and links mechanotransduction to neurological disorders. In *Sci. Adv* (Vol. 11). www.opentargets.org].
- Kumar, A., Mazzanti, M., Mistrik, M., Kosar, M., Beznoussenko, G. V., Mironov, A. A., Garrè, M., Parazzoli, D., Shivashankar, G. V., Scita, G., Bartek, J., & Foiani, M. (2014). ATR mediates a checkpoint at the nuclear envelope in response to mechanical stress. *Cell*, *158*(3), 633–646. <https://doi.org/10.1016/j.cell.2014.05.046>
- Lamm, N., Read, M. N., Nobis, M., Van Ly, D., Page, S. G., Masamsetti, V. P., Timpson, P., Biro, M., & Cesare, A. J. (2020). Nuclear F-actin counteracts nuclear deformation and promotes fork repair during replication stress. *Nature Cell Biology*, *22*(12), 1460–1470. <https://doi.org/10.1038/s41556-020-00605-6>
- Shao, X., Kawauchi, K., Shivashankar, G. V., & Bershady, A. D. (2015). Novel localization of formin mDia2: importin β -mediated delivery to and retention at the cytoplasmic side of the nuclear envelope. *Biology Open*, *4*(11), 1569–1575. <https://doi.org/10.1242/bio.013649>
- Wang, Y., Arjonen, A., Pouwels, J., Ta, H., Pausch, P., Bange, G., Engel, U., Pan, X., Fackler, O. T., Ivaska, J., & Grosse, R. (2015). Formin-like 2 Promotes β 1-Integrin Trafficking and Invasive Motility Downstream of PKC α . *Developmental Cell*, *34*, 1–9. <https://doi.org/https://doi.org/10.1016/j.devcel.2015.06.015>

Dear Robert,

Thank you for submitting a revised version of your manuscript. We have now received input from two of the original reviewers, who are generally satisfied with the revisions. There now remain only a few editorial points that need to be addressed before I can extend official acceptance of the manuscript:

1. Please submit up to five keywords.
2. Please make sure that the order of the sections in the manuscript is as follows: abstract, introduction, results, discussion, materials & methods, data availability section, acknowledgments, disclosure and competing interests statement, references, main figure legends, tables, expanded view figure legends.
3. CRediT has replaced the traditional author contributions section because it offers a systematic, machine-readable author contributions format that allows for more effective research assessment. Please remove the Authors Contributions from the manuscript and use the free text boxes beneath each contributing author's name in our online submission system to add specific details on the author's contribution. More information is available in our guide to authors.
4. Figure panel 3A is not mentioned in the manuscript text. Please add the corresponding callout.
5. Please rename "Data and materials availability" section into "Data availability".
6. Please remove movie legends from the manuscript text file and zip together with each movie file as a README text file. Further information is available here: <https://www.embopress.org/page/journal/14602075/authorguide#expandedview>
7. Please remove the Reagents and Tools Table from the manuscript text file and upload it as a separate file choosing the file type "Reagent Table".
8. In our standard image check, we noticed that images have been reused between figures 4A-EV3A and 4D-EV3A. If this is intentional, please clearly indicate this in the figure legends.
9. Our data editors have flagged the following issues in figure legends that need correcting:
 - Please provide the exact p values in the legends of figures 1B, D; 2B, D; 4B, C, E, F; 5E, F; 6D, 7D, F; EV2 H, EV4 F.
 - Please define the error bars in the legends of figures 8D, 9A-C.
 - Please define the scale bar for figure EV3 A.
 - Please define the white arrow heads in the legend of figure EV1 D.
10. Please submit a synopsis image with dimensions of 550 pixels width and 300-600 pixels height (jpeg or png format). You can either show a model or key data in the synopsis image.
11. Please remove the synopsis bullet points from the manuscript text file. I would like to propose some edits in the synopsis, manuscript title, and abstract. I have also written a short blurb that will accompany the title of your manuscript in our online table of contents. Please take a look at the proposed edits in the attached text file and let me know if any adjustments are necessary.

With best wishes,

leva

leva Gailite, PhD
Senior Scientific Editor
The EMBO Journal
Meyerhofstrasse 1
D-69117 Heidelberg
Tel: +4962218891309
i.gailite@embojournal.org

We realize that it is difficult to revise to a specific deadline. In the interest of protecting the conceptual advance provided by the work, we recommend a revision within 3 months (17th Nov 2025). Please discuss the revision progress ahead of this time with the editor if you require more time to complete the revisions.

Referee #2:

I am satisfied with the revisions. This work provides compelling evidence for a novel role for nuclear actin polymerization during cell migration.

Referee #3:

The authors have made efforts to address all reviewer comments and while not all of the experiments were successful, the main technical concerns have been addressed and the manuscript has improved and is ready for publication.

The authors addressed the remaining formatting issues.

Dear Robert,

Thank you very much for addressing the final editorial requests. I am now pleased to inform you that your manuscript has been accepted for publication in the EMBO Journal.

If you have any questions, please do not hesitate to contact the Editorial Office. Thank you for your contribution to The EMBO Journal, and congratulations with a nice study!

With best wishes,

Ieva
